# Observation of coordinated RNA folding events by systematic cotranscriptional RNA structure probing

Courtney E. Szyjka [1] & Eric J. Strobel [1] ✉

RNA begins to fold as it is transcribed by an RNA polymerase. Consequently, RNA folding is constrained by the direction and rate of transcription. Understanding how RNA folds into secondary and tertiary structures therefore requires methods for determining the structure of cotranscriptional folding intermediates. Cotranscriptional RNA chemical probing methods accomplish this by systematically probing the structure of nascent RNA that is displayed from an RNA polymerase. Here, we describe a concise, high-resolution cotranscriptional RNA chemical probing procedure called variable length Transcription Elongation Complex RNA structure probing (TECprobe-VL). We demonstrate the accuracy and resolution of TECprobe-VL by replicating and extending previous analyses of ZTP and fluoride riboswitch folding and mapping the folding pathway of a ppGpp-sensing riboswitch. In each system, we show that TECprobe-VL identifies coordinated cotranscriptional folding events that mediate transcription antitermination. Our findings establish TECprobe-VL as an accessible method for mapping cotranscriptional RNA folding pathways.

RNA begins to fold as it emerges from an RNA polymerase (RNAP) during transcription[1,2]. Consequently, the rate and 5' to 3' direction of transcription directly influence RNA folding, and nascent RNA structures can interact with RNAP to control transcription[3–5]. Although the role of transcription in RNA structure formation and the coordination of cotranscriptional processes has been established for decades[3,6,7], our ability to measure cotranscriptional RNA folding remained limited until recently. In the past several years, the development of methods that can monitor cotranscriptional RNA structure formation with high temporal and spatial resolution has rapidly advanced our ability to characterize cotranscriptional RNA folding mechanisms. These complementary approaches include applications of single-molecule force spectroscopy[8,9] and single-molecule FRET[10–13], which measure cotranscriptional RNA folding with high temporal resolution, and the application of high-throughput RNA chemical probing to map the structure of cotranscriptionally folded intermediate transcripts at nucleotide resolution[14–16].

High-throughput RNA chemical probing methods characterize the structure of complex RNA mixtures at nucleotide resolution and are compatible with diverse experimental conditions[17–19]. This experimental flexibility enabled the development of RNA chemical probing assays in which *E. coli* RNAP is distributed across template DNA so that cotranscriptionally folded intermediate transcripts can be chemically probed[14–16]. This approach systematically maps the structure of potential cotranscriptional RNA folding intermediates in the context of RNAP and identifies when secondary and tertiary structures can fold within an RNA folding pathway. Despite their utility for mapping RNA folding pathways, cotranscriptional RNA chemical probing experiments can be challenging to execute because existing protocols are relatively complex.

We have developed a concise, SHAPE-MaP-based[20] (selective 2'-hydroxyl acylation analyzed by primer extension and mutational profiling) procedure for multi-length cotranscriptional RNA chemical probing assays called TECprobe-VL (variable length Transcription Elongation Complex RNA structure probing) (Fig. 1). TECprobe-VL was designed with the specific objectives of standardizing cotranscriptional RNA structure probing assays and eliminating target-dependent

[1]Department of Biological Sciences, The University at Buffalo, Buffalo, NY 14260, USA. ✉e-mail: estrobel@buffalo.edu

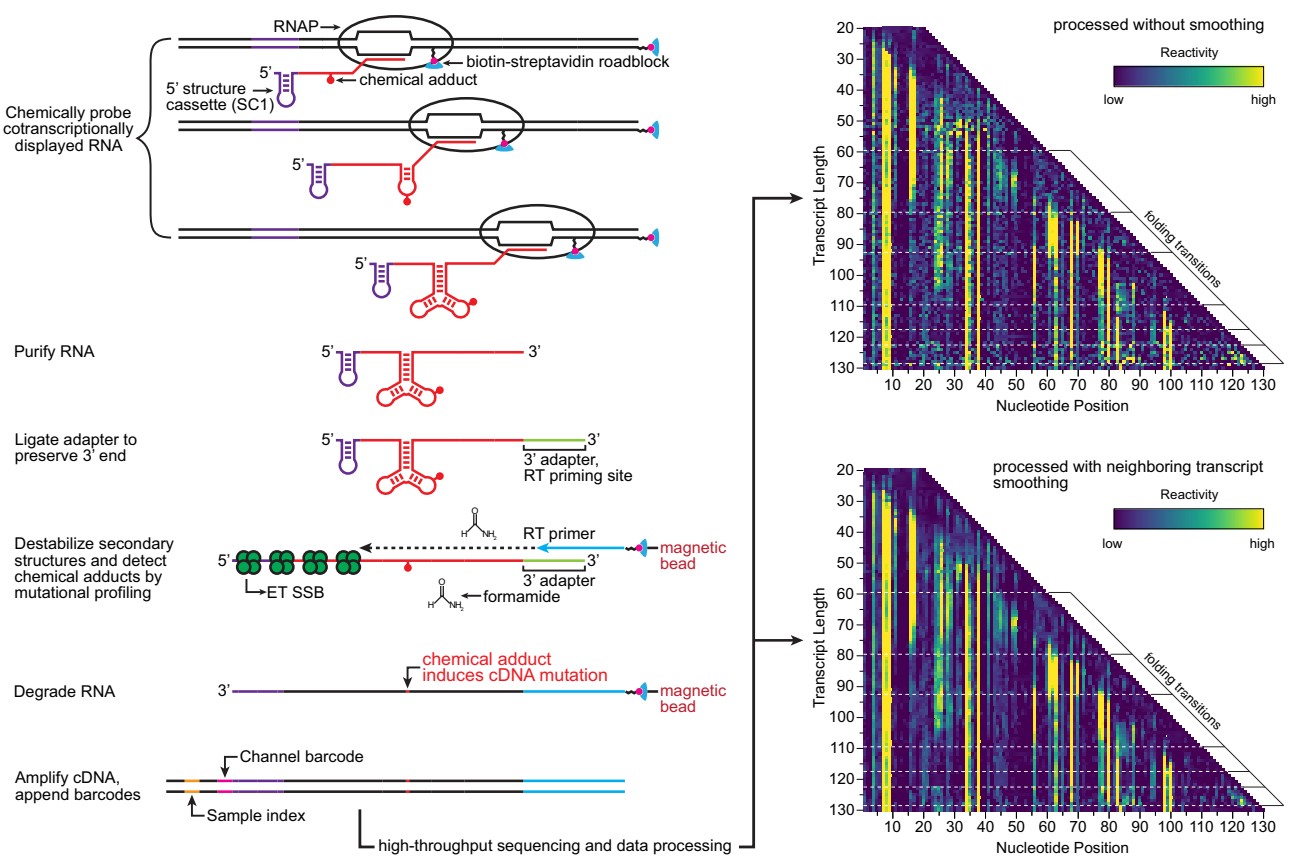

**Fig. 1 | Overview of TECprobe-VL.** Randomly biotinylated DNA templates are in vitro transcribed to generate cotranscriptionally folded intermediate transcripts displayed from *E. coli* RNAP. The cotranscriptionally displayed intermediate transcripts are chemically probed and purified. An adapter is ligated to the RNA 3' end and used to prime solid-phase error-prone reverse transcription. Full-length cDNA is amplified using primers that anneal to the adapter and the 5' structure cassette. Following high-throughput sequencing, data are processed using fastp[78], TECtools, and ShapeMapper2[79] either with or without neighboring transcript smoothing. SC1 structure cassette 1, RT reverse transcription, ET SSB extreme thermostable single-stranded DNA binding protein.

variability in experiment outcomes. At the experimental level, this is accomplished through a streamlined procedure that yields sequencing libraries with a ~90% alignment rate and guarantees even sequencing coverage across the length of each intermediate transcript. At the computational level, we implemented a data smoothing procedure to overcome experimentally irresolvable biases in the sequencing coverage of intermediate transcripts. We validated TECprobe-VL by repeating and extending previous analyses of the *Clostridium beijerinckii* (*C. beijerinckii*, *Cbe*) *pfl* ZTP[21] and *Bacillus cereus* (*B. cereus*, *Bce*) *crcB*[22] riboswitches. In addition to replicating all prior observations, TECprobe-VL resolved biologically relevant folding events that were not previously detectable. We then used TECprobe-VL to map the folding pathway of a ppGpp-sensing riboswitch[23] from *Clostridiales bacterium oral taxon* 876 str. F0540 (*C. bacterium*, *Cba*). Our analysis determined that (i) the *Cba* ppGpp riboswitch folds using a branched pathway in which mutually exclusive intermediate structures converge to form the ppGpp aptamer, and (ii) ppGpp binding coordinates extensive long-range contacts between ppGpp aptamer subdomains. Together, these advances establish TECprobe-VL as a high-performance and broadly accessible cotranscriptional RNA structure probing method.

## Results
### Optimization and benchmarking of TECprobe-VL
TECprobe-VL is designed to simplify and standardize cotranscriptional RNA structure probing experiments. Like Cotranscriptional SHAPE-seq, TECprobe-VL experiments begin by distributing TECs across template DNA in a single-round in vitro transcription reaction using biotin-streptavidin roadblocks, which halt RNAP ~10 nts upstream of the biotin site, so that cotranscriptionally displayed intermediate transcripts can be chemically probed[15]. Internal biotin modifications are incorporated into template DNA by including biotin-11-dNTPs during PCR and a terminal biotin modification that prevents any run-off transcription with >99% efficiency[24] is included in the reverse primer used for PCR. Internal biotin-streptavidin roadblocks halt transcription more efficiently when they are positioned in the template DNA strand (>80%) than in the nontemplate DNA strand (~30%)[15]. Consequently, DNA templates that are structured as described above enrich for intermediate transcripts from the transcription start site until ~10 nucleotides before the segment of template DNA that is composed of the reverse primer and at a cluster of positions ~10 nucleotides upstream of the terminal biotin-streptavidin complex (Fig. 2b-d, Supplementary Fig. 1a-c, top plots).

TECprobe-VL primarily simplifies cotranscriptional RNA structure probing experiments by including an established 5' structure cassette[25] (referred to here as SC1) upstream of the target RNA sequence and detecting chemical adducts by mutational profiling using the SHAPE-MaP[20] strategy (Fig. 1). Mutational profiling uses error-prone reverse transcription to encode the location of RNA chemical adducts, which increase the error rate of reverse transcriptase, as mutations in cDNA[20]. Because SHAPE-MaP can encode reactivity information in full-length cDNA molecules, the inclusion of a 5' structure cassette upstream of the target RNA facilitates direct amplification of cDNA[20]. In contrast, the original Cotranscriptional SHAPE-Seq procedure required the ligation of an adapter to the cDNA 3' end because chemical adducts were detected as truncated cDNA[14,15]. Performing this ligation using the

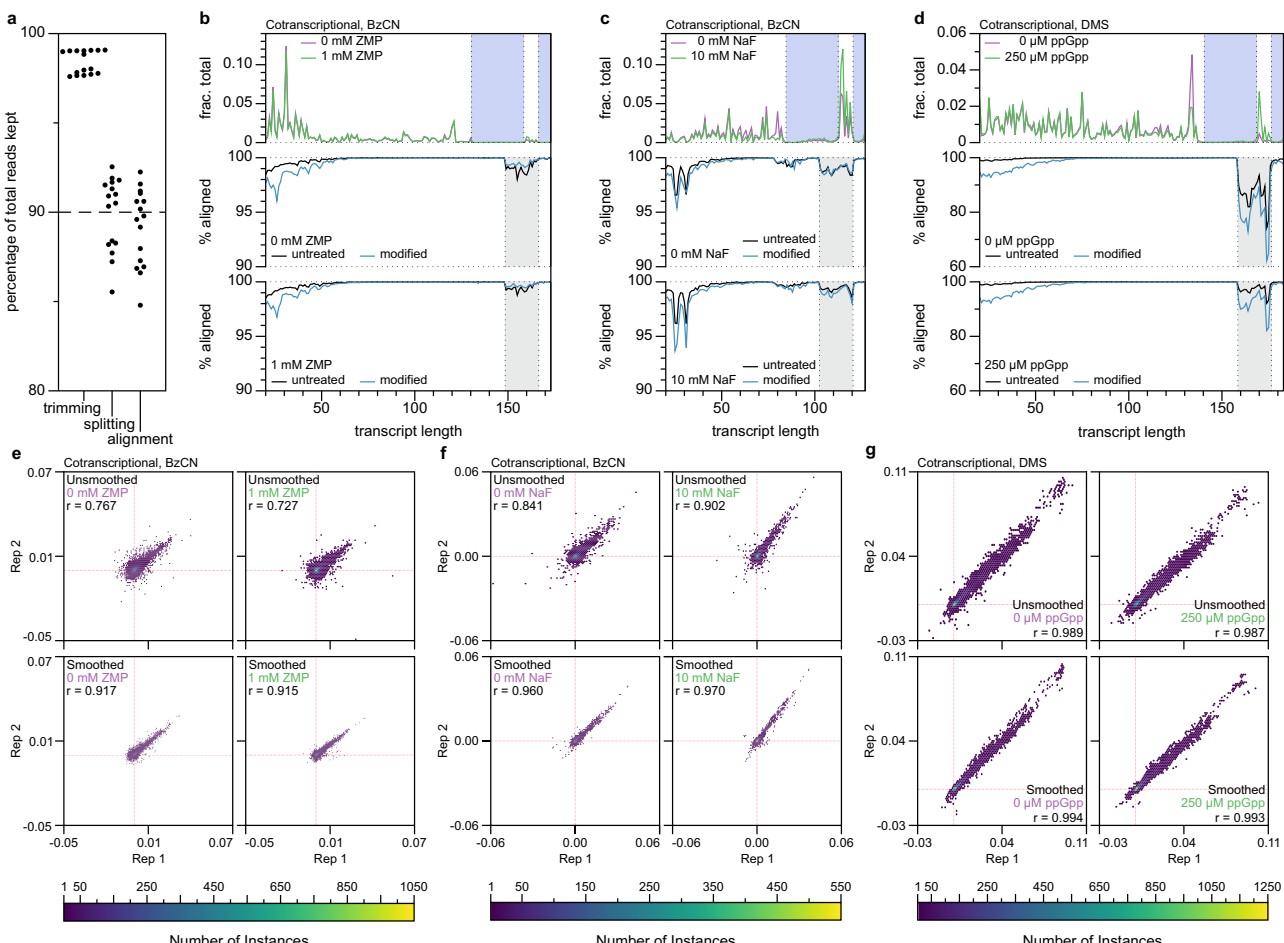

**Fig. 2 | TECprobe-VL performance benchmarks. a** Percentage of total reads kept after adapter trimming, splitting fastq files by transcript length and channel barcode, and sequencing read alignment. **b–d** Plots showing the fraction of aligned reads that mapped to each transcript length (top) and the percentage of split reads that aligned to each transcript length (middle, bottom) for the *Cbe pfl* ZTP riboswitch (BzCN) (**b**), the *Bce crcB* fluoride riboswitch (BzCN) (**c**), and the *Cba* ppGpp riboswitch (DMS) (**d**) datasets. In the top plots, data from minus and plus ligand samples are purple and green, respectively. The blue shading in the top plot indicates transcripts that were not enriched by biotin-streptavidin roadblocks. The grey shading in the middle and lower plots indicates transcripts in which alignment was lower due to the presence of a nucleic acid species that is kept during fastq splitting but does not align. **e-g**, Hexbin plots comparing the reactivity of replicates for the *Cbe pfl* ZTP (BzCN) (**e**), *Bce crcB* fluoride (BzCN) (**f**), and *Cba* ppGpp (DMS) (**g**) riboswitches for both unsmoothed (top) and smoothed (bottom) datasets. Data were plotted with a grid size of 75 by 75 hexagons, and the depth of overlapping data points is indicated by the heatmap. Red dashed lines indicate the position of 0 for each axis. Source data are provided as a Source Data file. BzCN benzoyl cyanide, DMS dimethyl sulfate.

ssDNA ligase CircLigase, which is prone to sequence bias[26,27], can distort the reactivity measurement and cause the formation of adapter dimer, which reduces the number of usable sequencing reads. A procedure for appending an adapter to the cDNA 3′ end using a splinted ligation, which does not have nucleotide bias, was developed for high-throughput RNA structure probing experiments[26,28]. However, this approach can still cause the formation of adapter dimer. In the original Cotranscriptional SHAPE-Seq procedure, the removal of adapter dimer by size selection also depleted short cDNA products, resulting in poor resolution of 3′-proximal RNA structures. Because this size selection procedure attempted to retain short cDNA products, the removal of adapter dimer was often incomplete, which caused alignment rates of 20% to 60%, depending on the abundance of adapter dimer in the sequencing library. Consequently, eliminating the need to ligate an adapter to the cDNA 3′ end resolves every limitation of the original Cotranscriptional SHAPE-Seq procedure that was caused by adapter dimer: TECprobe-VL libraries have a ~90% alignment rate (Fig. 2a), and resolve 3′-proximal structures that were not detectable previously (as is shown for the ZTP and fluoride riboswitches below).

In addition to eliminating the need for cDNA ligation, we optimized several steps of the TECprobe-VL library prep. First, ligation of

an RNA 3′ adapter is performed using the reverse complement of the Illumina TruSeq Small RNA Kit RA5 adapter so that Read 1 begins with high-complexity sequence, which is necessary for cluster identification during Illumina sequencing. Second, after the RNA 3′ adapter ligation, RNA is purified using solid-phase reversible immobilization (SPRI) to remove most excess adapter prior to reverse transcription (Supplementary Figs. 2 and 4a). Third, reverse transcription is performed as a solid-phase reaction to simplify cDNA library purification (Fig. 1). Trace amounts of excess RNA 3′ adapter in the solid-phase reverse transcription reaction caused the formation of an RT primer product due to template switching (Supplementary Figs. 3a, b and 4a). However, this product was not amplifiable and therefore had no effect on the procedure (Supplementary Fig. 3c). Fourth, reverse transcription is performed in the presence of extreme thermostable single-stranded DNA binding protein (ET SSB) and formamide to eliminate RT primer dimer and promote full-length cDNA synthesis (Fig. 1 and Supplementary Fig. 4b). Formamide and SSB are commonly used to improve primer specificity and relieve secondary structure during PCR and have been used for the same purposes during reverse transcription[29–32]. ET SSB eliminated RT primer dimer, but neither ET SSB nor formamide affected the efficiency at which a ZTP riboswitch RNA was reverse

transcribed (Supplementary Fig. 4b). Because the inclusion of up to 5% formamide did not inhibit reverse transcription, we chose to include 2% formamide in the reaction, which may relieve secondary structure in some RNA targets. In combination, structuring full-length cDNA as an amplicon and the protocol optimizations described above enabled us to reduce the scale of the TECprobe-VL in vitro transcription reaction by 10-fold relative to the original Cotranscriptional SHAPE-Seq protocol.

The remaining limitation of cotranscriptional RNA structure probing procedures is that sequence and structural biases from the RNA 3' adapter ligation can cause uneven representation of intermediate transcripts in the sequencing library[15] (Fig. 2b-d and Supplementary Fig. 1a-c, upper plots). Because there is unlikely to be a general experimental solution to this limitation, we implemented a computational solution. Our approach is based on the observation that neighboring transcripts tend to have similar reactivity patterns and background mutation rates (Supplementary Fig. 5). Therefore, concatenating sequencing reads from transcripts $n-1$, $n$, and $n+1$ in overlapping windows across a cotranscriptional RNA structure probing data set is expected to preserve transcript-length-dependent reactivity trends while boosting the effective sequencing depth for each length $n$. The use of neighboring transcript smoothing improved the correlation between replicate data sets, particularly when BzCN was used (Fig. 2e-g and Supplementary Fig. 1d-f). Comparisons of unsmoothed and smoothed data sets for each riboswitch RNA assessed in this work are shown in Supplementary Fig. 6.

### Validation: The *C. beijerinckii pfl* ZTP riboswitch

Strobel et al. previously used Cotranscriptional SHAPE-Seq to map the folding pathway of the *Cbe pfl* ZTP riboswitch[33], which controls gene expression in response to the purine metabolites ZMP and ZTP using a transcription antitermination mechanism[21] (Supplementary Fig. 7). ZTP riboswitch aptamers comprise two subdomains (P1/P2 and P3) that are associated by a pseudoknot and make extensive ZMP-dependent tertiary contacts[21,34–36] (Fig. 3a, h). ZMP binding is mediated by nucleotides in L3 and by $Mg^{2+}$-mediated recognition of the Z nucleobase[34–36]. It was previously determined that the *pfl* riboswitch folding pathway involves the formation of a non-native intermediate hairpin (IH1) and that ZMP binding coordinates tertiary contacts between the P1/P2 and P3 subdomains that render P3 resistant to terminator hairpin nucleation[33]. However, most of the evidence supporting these conclusions was from 5'-proximal reactivity signatures due to the limited resolution of 3'-proximal nucleotides. To validate TECprobe-VL, we confirmed that the SC1 leader did not meaningfully impact *pfl* ZTP riboswitch function, repeated the analysis of *pfl* riboswitch folding using the SHAPE probe benzoyl cyanide (BzCN)[37,38], and extended this analysis to include dimethyl sulfate (DMS) probing[39,40] (Fig. 3b-d and Supplementary Fig. 8a, b).

*Pfl* riboswitch folding begins with the rearrangement of IH1 (nts 10–23) to form the P1/P2 subdomain[33] (Fig. 3a). P1 folding was detected as decreased reactivity in the IH1 loop and the downstream P1 stem from transcript ~67 to ~80, as the downstream P1 stem emerges from RNAP (Fig. 3a, e and Supplementary Fig. 8c). P2 folding was detected at transcript ~60 as decreased P2 reactivity when the entire P2 hairpin has emerged from RNAP (Fig. 3e and Supplementary Fig. 8d). However, the BzCN reactivity of the upstream P2 stem increases when IH1 refolds into P1 at transcript ~74, suggesting that the flexibility of P2 increases in the context of the folded P1/P2 subdomain (Fig. 3e). In previous data, the IH1-to-P1 folding transition was detectable as decreased IH1 loop reactivity[17] (Supplementary Fig. 9a). However, with the exception of U49, the reactivity of downstream P1 nucleotides was constitutively low when the IH1-to-P1 folding transition occurs, which prevented the direct observation of P1 folding[33] (Supplementary Fig. 9b, d). Neither P2 folding nor the increase in P2 flexibility that occurs upon P1 folding were previously detectable (Supplementary Fig. 9c).

The P1/P2 and P3 subdomains are connected by a linker composed of a hairpin and an unstructured region[21] (Fig. 3a). Folding of the linker hairpin was detected as decreased reactivity within the upstream and downstream segments of its stem at transcript 93, when the entire hairpin has emerged from RNAP (Fig. 3g and Supplementary Fig. 8f). Linker hairpin folding was detectable in previous data (Supplementary Fig. 9e).

The *pfl* riboswitch pseudoknot comprises base pairs between nucleotides 23–27 in J1/2 and nucleotides 89–93 in L3 (nts 89–93)[21] (Fig. 3a). ZMP-independent pseudoknot folding was detected as decreased reactivity in J1/2 from transcript ~104 to ~110 when nucleotides 89–93 emerge from RNAP (Fig. 3f and Supplementary Fig. 8e). Pseudoknot folding is coordinated with decreased BzCN reactivity in P2, in the terminal base pair of P1, and in P1-proximal unpaired nucleotides within the inter-subdomain linker (Fig. 3e). This indicates that, as expected, pseudoknot folding drives global changes in aptamer structure. While pseudoknot folding was detected in previous data, the coordinated reactivity changes in P1 and the inter-subdomain linker were not[33] (Supplementary Figs. 9d and 10a).

P3 hairpin folding was not clearly observable in previous data due to poor resolution of 3'-proximal nucleotides[33]. Consequently, ZMP binding was inferred from reactivity changes in the P1/P2 subdomain even though ZMP primarily interacts with L3 (Fig. 3h). In contrast, the current TECprobe-VL data provide a high-resolution map of P3 folding and direct evidence for ZMP binding. P3 folding was detected as decreased reactivity in the upstream P3 stem from transcript 114 to 118, after the downstream P3 stem has emerged from RNAP (Fig. 3f and Supplementary Fig. 8e). In previous data, the decrease in upstream P3 stem reactivity that occurs upon P3 folding was observed clearly for G84 but was not detected at C85 and C86 at the time of analysis because the reactivity at these positions is 10-fold lower than that of G84 (Supplementary Fig. 10b). Consequently, it was not previously possible to confidently interpret decreased G84 reactivity as P3 folding. In coordination with P3 folding, a ZMP-dependent decrease in BzCN reactivity occurred at U94, which forms two hydrogen bonds with the Z nucleobase, and G95, which stacks with the Z nucleobase[34–36] (Fig. 3f). In agreement with the observation that ZMP binding renders P3 resistant to terminator hairpin nucleation[33], the BzCN reactivity of the terminator nucleation point (C98) was lower with ZMP (Fig. 3f). These previously undetectable[33] reactivity signatures are direct evidence for ZMP binding and the mechanism of ZMP-mediated transcription antitermination (Supplementary Fig. 10c, d).

P3 folding is coordinated with extensive ZMP-dependent reactivity changes in the P1/P2 subdomain, most of which were observed previously[33] (Supplementary Fig. 11a, Supplementary Fig. 12). First, ZMP-dependent decreased reactivity at A34 and A38 corresponds to reduced L2 flexibility and the formation of a conserved A-minor motif[34–36], respectively (Supplementary Fig. 11b, c). Second, ZMP-dependent decreased reactivity in upstream and downstream P1 nucleotides corresponds to the stabilization of noncanonical P1 base pairs, which most likely results from the formation of a ribose zipper between P1 and P3[33–36] (Supplementary Fig. 11d-g). Consistent with the formation of a Hoogsteen/Hoogsteen base pair between G19 and A47[34–36], which leaves the Watson-Crick face of A47 unpaired, A47 remained reactive to DMS regardless of whether ZMP was present (Supplementary Fig. 11h). Third, a ZMP-dependent decrease in the BzCN reactivity of A29 that is most likely due to stacking with the J1/2-L3 pseudoknot was previously observed[33]. In the current data, this was not detectable by BzCN probing. However, increased DMS reactivity at A29 suggests that ZMP binding places A29 in an unpaired conformation (Supplementary Fig. 11i).

Formation of the terminator hairpin requires disruption of both P3 and the pseudoknot[21] (Fig. 3a). Without ZMP, terminator hairpin nucleation is detected at transcript 123 as increased reactivity in J1/2

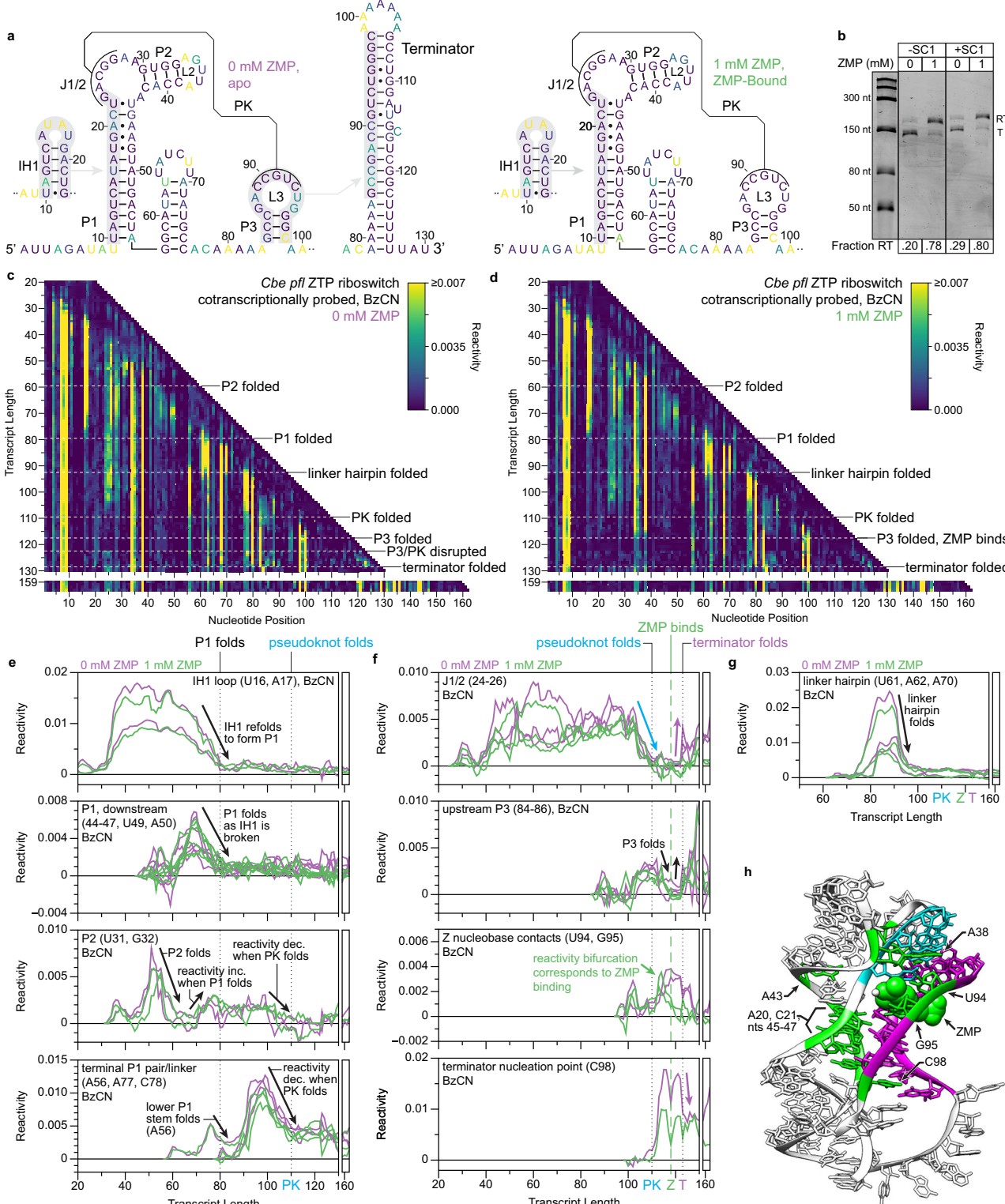

**Fig. 3 | Cotranscriptional folding transitions of the *C. beijerinckii pfl* ZTP riboswitch. a** Secondary structures of *pfl* ZTP riboswitch folding intermediates colored by reactivity (IH1: BzCN, 41 nt; Apo/ZMP-bound aptamer: BzCN, 120 nt; terminator: DMS, 130 nt). **b** Single-round transcription termination assays for the *pfl* ZTP riboswitch with and without SC1. Fraction readthrough is the average of two independent replicates. An uncropped gel image is provided in Supplementary Fig. 24. **c, d** TECprobe-VL BzCN reactivity matrices for the *pfl* ZTP riboswitch with 0 mM and 1 mM ZMP. Transcripts 131–158, which were not enriched, are excluded. Data are from two independent replicates that were concatenated and analyzed together. Reactivity is shown as background-subtracted mutation rate. **e-g**

Transcript length-dependent reactivity changes showing IH1-to-P1 and P2 formation (**e**), pseudoknot formation/disruption and ZMP binding (**f**), and linker hairpin formation (**g**) folding transitions. Data from 0 mM and 1 mM ZMP samples are purple and green, respectively. Vertical dotted and dashed lines mark when the indicated folding transitions occur. **h** *Thermosinus carboxydivorans pfl* riboswitch structure (PDB 4ZNP)[35]. P3 (magenta), J1/2 PK (cyan), and ZMP-responsive nucleotides (green) are highlighted. Source data are provided as a Source Data file. BzCN benzoyl cyanide, DMS dimethyl sulfate, PK pseudoknot, SC1 structure cassette 1, RT readthrough, T terminated.

and the upstream P3 stem, which become unpaired when the J1/2-L3 pseudoknot and P3 are broken (Fig. 3f and Supplementary Fig. 8e). At transcript 129, decreased DMS reactivity in the upstream P3 stem is consistent with complete terminator hairpin folding (Supplementary Fig. 8e). With ZMP, the BzCN reactivity of the upstream P3 stem also increases when the terminator hairpin can nucleate at transcript 123, however there is no corresponding increase in the BzCN or DMS reactivity of J1/2, indicating that the pseudoknot is intact (Fig. 3f and Supplementary Fig. 8e). Notably, the DMS reactivity of the upstream P3 stem remains low in the presence of ZMP, which suggests that P3 remains intact despite its increased flexibility (Supplementary Fig. 8e). In previous data, pseudoknot disruption was detectable as increased reactivity in J1/2, but the effect of terminator folding on P3 was not interpretable (Supplementary Fig. 10a, b).

Visualization of terminator readthrough transcripts was difficult using Cotranscriptional SHAPE-Seq because the ZTP riboswitch terminator hairpin is a robust structural barrier to reverse transcription in the conditions used to detect modification sites as cDNA truncions[33]. The reverse transcription conditions used for TECprobe-VL facilitated nearly complete terminator bypass and enabled high-resolution data to be obtained for full-length ZTP riboswitch transcripts (Supplementary Fig. 4b). As expected, transcripts from terminally roadblocked TECs adopted alternate structures depending on whether ZMP was included in the reaction. Without ZMP, increased reactivity at J1/2 indicates that the pseudoknot was disrupted (Fig. 3f and Supplementary Fig. 8e, transcripts 159 to 162). With ZMP, low reactivity at J1/2 indicates that the pseudoknot remained intact, and the persistence of ZMP-dependent reactivity signatures indicates that ZMP remained bound to the aptamer (Fig. 3f, and Supplementary Figs. 8e and 11). BzCN probing of full-length ZTP riboswitch transcripts using a variation of the TECprobe procedure designed to target single transcript lengths (TECprobe-SL) yielded identical results (Supplementary Fig. 13). Like the upstream terminator stem, which comprises the P3 hairpin, the downstream terminator stem can form an alternative hairpin that spans nucleotides 101 to 127 (Supplementary Fig. 8g). The clearest evidence for the formation of this alternative structure is that A and C nucleotides within its loop are reactive to DMS in the presence of ZMP, when the ZMP-bound aptamer is intact, but not in the absence of ZMP, when the terminator hairpin has folded (Supplementary Fig. 8g). Once folded, this structure precludes terminator hairpin nucleation, which may contribute to the persistence of the ZMP-bound aptamer.

In sum, TECprobe-VL detected all *pfl* ZTP riboswitch folding events that were detected previously by Cotranscriptional SHAPE-Seq[33] and facilitated clear visualization of P3 subdomain folding, ZMP-binding, and terminator readthrough transcripts, which was not previously possible.

## Validation: the *B. cereus crcB* fluoride riboswitch

Watters, Strobel et al. previously used Cotranscriptional SHAPE-Seq to map the folding pathway of the *Bce crcB* fluoride riboswitch[14], which controls gene expression in response to fluoride using a transcription antitermination mechanism[22]. The fluoride aptamer comprises an H-type pseudoknot and long-range reversed Watson-Crick and Hoogsteen A-U base pairs, and coordinates three $Mg^{2+}$ ions which in turn coordinate the fluoride anion[41] (Fig. 4a, h). The previous analysis of *crcB* riboswitch folding found that fluoride binding stabilizes the pseudoknot, promotes formation of the reversed Watson-Crick A-U pair, reduces the overall flexibility of the aptamer, delays terminator hairpin nucleation, and blocks propagation of terminator base pairs[14,15] (Supplementary Fig. 14). To further validate TECprobe-VL, we confirmed that the SC1 leader did not meaningfully impact *crcB* fluoride riboswitch function, repeated the analysis of the *crcB* riboswitch using BzCN, and extended this analysis to include DMS probing (Fig. 4b-d and Supplementary Fig. 15a, b).

The *crcB* riboswitch pseudoknot comprises base pairs between nucleotides 12–17 of L1 and nucleotides 42–47[22] (Fig. 4a). Pseudoknot folding was detected as decreased reactivity in both L1 and nucleotides 43–47 beginning at transcript ~59, when nucleotides 42–47 are emerging from RNAP (Fig. 4e). With fluoride, pseudoknot reactivity decreases abruptly from transcript ~59 to ~61 and reaches a lower minimum reactivity at transcript ~70, when the pseudoknot has fully folded, than in the absence of fluoride (Fig. 4e). This is consistent with the snap-lock model for fluoride aptamer stabilization, in which the apo aptamer explores multiple states dynamically, but the holo aptamer exists in a stably docked state[42]. Previous data relied solely on reactivity changes within L1 to track pseudoknot folding because of poor resolution in 3′-proximal nucleotides[14,15] (Supplementary Fig. 16a, b).

Formation of the *crcB* aptamer pseudoknot is coordinated with numerous fluoride-dependent reactivity changes (Fig. 4g). First, the reactivity of A10 and U11 within L1, which are not involved in pseudoknot base pairs, increases upon pseudoknot folding when fluoride is absent but decreases when fluoride is present (Fig. 4f). The lower reactivity of A10 and U11 in the presence of fluoride most likely corresponds to the formation of a reversed Watson-Crick pair between A10 and U38[41] which extends the P3 stack through U11 (Fig. 4h). Consistent with this interpretation, the reactivity of U38 also decreases upon pseudoknot folding when fluoride is present (Fig. 4f). Formation of the A10-U38 base pair was detected in previous data[14,15] (Supplementary Fig. 16c, d). Second, A40 undergoes a fluoride-dependent decrease in BzCN reactivity and a fluoride-dependent increase DMS reactivity (Fig. 4f). This is consistent with the formation of a Hoogsteen A-U pair with U48[41], known as the linchpin base-pair, which locks the fluoride-bound aptamer into a stably docked state[43] (Fig. 4h). In addition, the reactivity of A49 and C50, which stack with the linchpin base pair[41], decreases upon pseudoknot folding in the presence of fluoride (Fig. 4f, h). In previous data, no ligand-dependent change in the reactivity of A40 was detected, and reactivity changes that occurred at A49 were not clearly coordinated with pseudoknot or terminator folding as they are in the current data[14,15] (Supplementary Fig. 16e, f). Third, within J1/3, the reactivity of nucleotides A22 and U23 increases in response to fluoride binding, whereas the reactivity of nucleotides A24 and A25 decreases (Supplementary Fig. 15c). Similar reactivity trends were observed within J1/3 in previous data but were less obvious[14,15] (Supplementary Fig. 17a-c). Fourth, a fluoride-dependent decrease in the DMS reactivity of P3 indicates that P3 is more stably base-paired in the fluoride-bound aptamer (Supplementary Fig. 15c).

It was previously observed that the fluoride-bound *crcB* aptamer delays terminator hairpin nucleation until after RNAP has bypassed the termination sites (+80 to +82) and blocks propagation of terminator base pairs when nucleation eventually occurs[14,15] (Supplementary Fig. 17d-f). Both fluoride-dependent delayed terminator hairpin formation (Supplementary Fig. 15d, e) and inhibition of terminator base pair propagation (Fig. 4e, transcripts 113–116) were detectable in the current TECprobe-VL data. In the presence of fluoride, the reactivity of downstream terminator stem nucleotides that cannot form terminator base pairs when the fluoride aptamer is intact (nts 65–75) remained low (Fig. 4c, d, Supplementary Fig. 15a, b). One possible explanation for the low reactivity of these nucleotides is that blocking terminator base pair propagation may allow unpaired terminator hairpin nucleotides to form an alternative structure (Fig. 4a, antiterminated structure). As described above for the ZTP riboswitch, this alternative hairpin could contribute to the persistence of the fluoride-bound aptamer by precluding terminator base pair propagation.

In sum, TECprobe-VL detected all *crcB* fluoride riboswitch folding events that were observed previously by Cotranscriptional SHAPE-Seq[14,15], and detected several signatures of linchpin base pair formation, which was not previously possible.

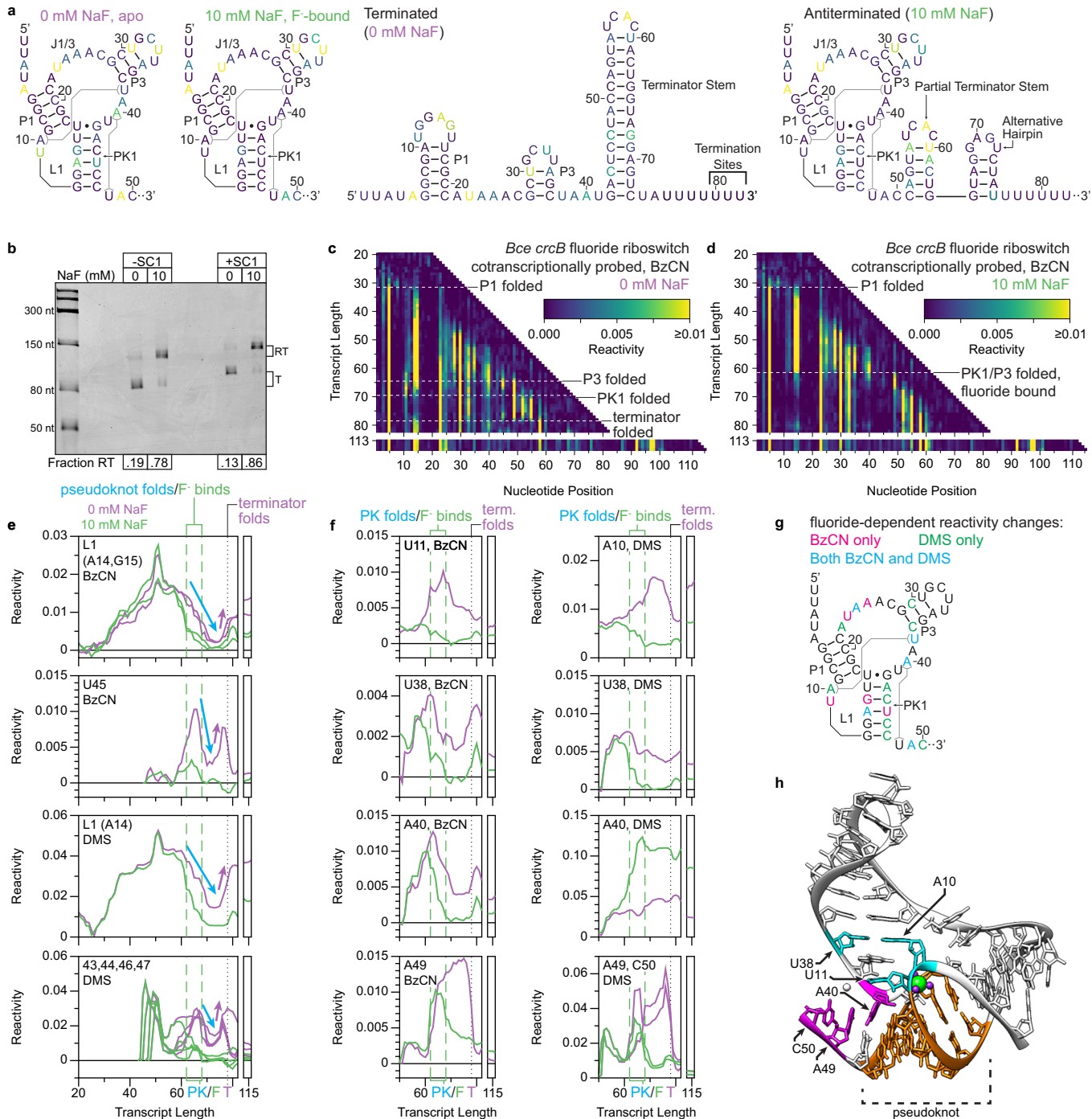

**Fig. 4 | Cotranscriptional folding transitions of the *B. cereus crcB* fluoride riboswitch. a** Secondary structures of *crcB* fluoride riboswitch folding intermediates colored by BzCN reactivity (apo/fluoride-bound aptamer: 68 nt; terminated: 82 nt, antiterminated 114 nt). **b** Single-round transcription termination assays for the *crcB* fluoride riboswitch with or without SC1. Fraction readthrough is the average of two independent replicates. An uncropped gel image is provided in Supplementary Fig. 24 **c, d**, TECprobe-VL BzCN reactivity matrices for the *crcB* fluoride riboswitch with 0 mM and 10 mM NaF. Transcripts 83–112, which were not enriched, are excluded. Data are from two independent replicates that were concatenated and analyzed together. Reactivity is shown as background-subtracted mutation rate. **e, f** Transcript length-dependent reactivity changes showing pseudoknot formation/disruption (**e**) and

long-range base pair formation (**f**) folding transitions. DMS data are from Supplementary Fig. 15a, b. Data from 0 mM and 10 mM NaF samples are purple and green, respectively. Vertical dotted and dashed lines mark when the indicated folding transitions occur. **g** Secondary structure of the *crcB* fluoride aptamer colored to show nucleotides that undergo fluoride-dependent reactivity changes observed using BzCN, DMS, or both probes. **h** *Thermotoga petrophila* fluoride riboswitch structure (PDB [4ENC])[41]. Pseudoknot (orange), A10-U38 base-pair-associated (cyan), and A40-U48 linchpin base-pair-associated (magenta) nucleotides are highlighted. F⁻ and Mg²⁺ are green and purple, respectively. Source data are provided as a Source Data file. BzCN benzoyl cyanide, PK pseudoknot, SC1 structure cassette 1, RT readthrough, T terminated, DMS dimethyl sulfate.

## Analysis of ppGpp riboswitch folding using TECprobe-VL

ppGpp riboswitches regulate the expression of genes associated with branched-chain amino acid biosynthesis and transport in response to the alarmone ppGpp[23] and are a member of the *ykkC* aptamer family,

which also includes guanidine[44], phosphoribosyl pyrophosphate[45], and nucleoside diphosphate[46] riboswitches (Fig. 5a, b). The ppGpp aptamer comprises two helical stacks (P2-P3 and P1-P4) that are connected by a four-way junction and are associated by long-range contacts

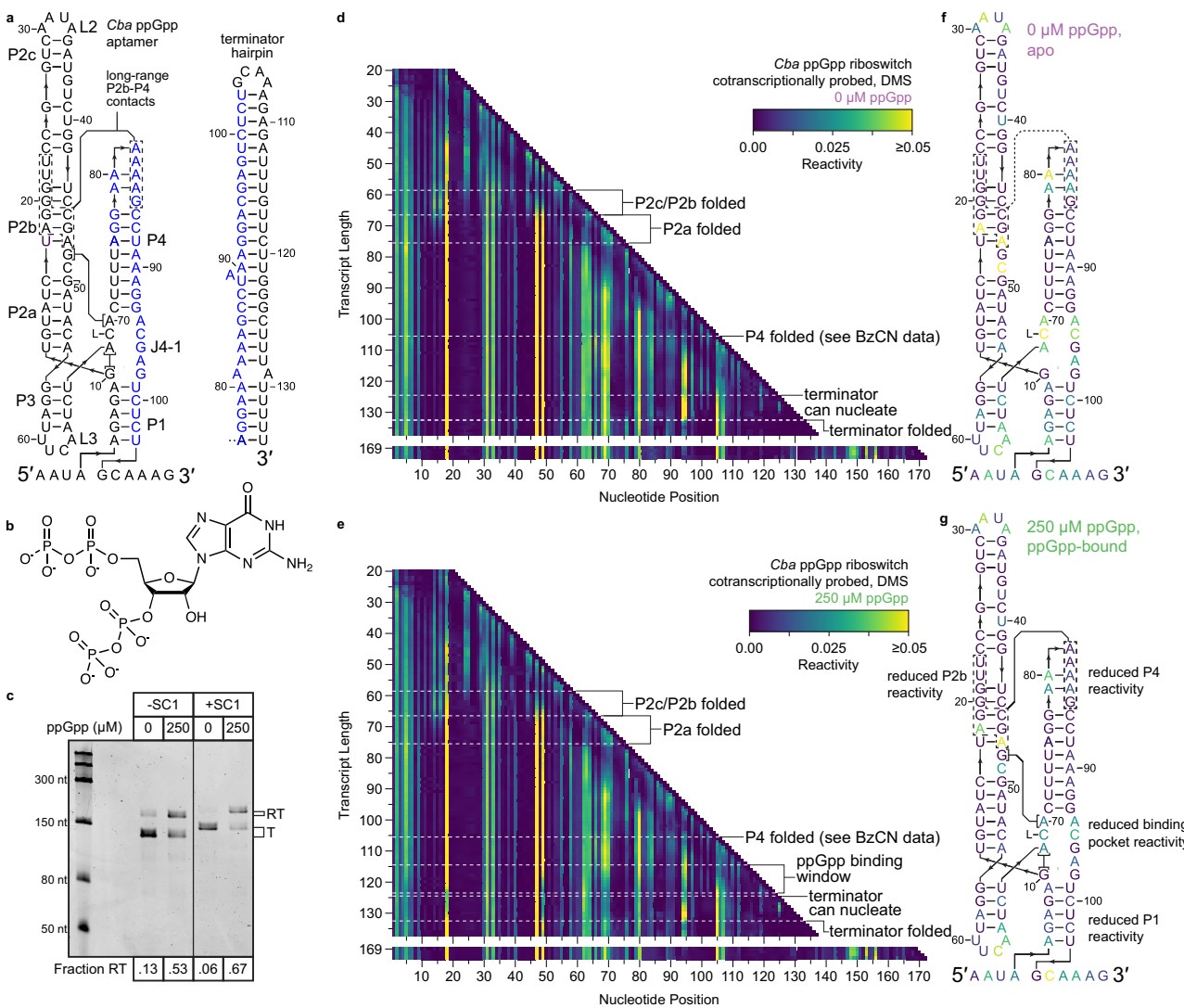

**Fig. 5 | Cotranscriptional chemical probing of the *C. bacterium* ppGpp riboswitch. a** Secondary structures of the *Cba* ppGpp aptamer (left) and terminator hairpin (right). Nucleotides present in both the aptamer and terminator hairpin are colored blue, interacting P4 and P2b nucleotides are shown by dashed boxes, and the nucleobase of ppGpp is shown as L. **b** Chemical structure of ppGpp. **c**, Single-round transcription termination assays for the *Cba* ppGpp riboswitch with or without SC1. Fraction readthrough is the average of four independent replicates. An uncropped gel image is provided in Supplementary Fig. 24. **d, e** TECprobe-VL DMS reactivity matrices for the *Cba* ppGpp riboswitch with 0 μM and 250 μM ppGpp. Transcripts 138–168, which were not enriched, are excluded. Data are from two independent replicates that were concatenated and analyzed together. Reactivity is shown as background-subtracted mutation rate. **f, g** *Cba* ppGpp aptamer secondary structures colored by DMS reactivity from transcript 123 in the absence (**f**) and presence (**g**) of ppGpp. Source data are provided as a Source Data file. SC1 structure cassette 1, RT readthrough, T terminated, DMS dimethyl sulfate, BzCN benzoyl cyanide.

between P2 and P4[47,48]. ppGpp recognition is mediated primarily by the P1-P4 helix and involves direct recognition of the G nucleobase by a conserved C in J3-4[47,48]. While the overall architecture of ppGpp aptamers and the mechanism of ligand recognition have been established, the mechanism of ppGpp aptamer folding and the ppGpp-dependence of aptamer structures remain unresolved. As was the case for the ZTP and fluoride riboswitches above, the SC1 leader hairpin did not meaningfully impact the function of the *Cba* ppGpp riboswitch (Fig. 5c). BzCN and DMS probing of ppGpp riboswitch folding intermediates using TECprobe-VL identified several transient structures that occur early in the ppGpp aptamer folding pathway and established which elements of the holo ppGpp aptamer structure are ligand-dependent (Fig. 5d-g and Supplementary Fig. 18a, b).

### Non-native folding intermediates precede P2 folding

The P2 helix of the *Cba* ppGpp aptamer spans nucleotides 11–55 (Fig. 5a). Consequently, P2 cannot fold completely until the nascent

transcript is at least 65–70 nt long, and most or all P2 nucleotides have emerged from RNAP. The clearest delineation of early ppGpp aptamer folding is observed when intermediate transcripts are equilibrated prior to BzCN probing, which reduces structural heterogeneity and precisely defines when transient structures become thermo-dynamically favorable (Supplementary Fig. 19a, b). Two intermediate structures were identified: The first intermediate hairpin (IH1) comprises base pairs between nucleotides 7–9 and 17–15; the IH1 stem is non-reactive, whereas nucleotides U11 and A14 of the IH1 loop are reactive to BzCN (Supplementary Fig. 19c). The second intermediate hairpin (IH2) comprises base pairs between nucleotides 7–21 and 40–28 with two asymmetrical internal bulges and is mutually exclusive with IH1 (Supplementary Fig. 19d). The primary evidence for IH2 is that the reactivity of A14, which is paired in IH2 but not in IH1, decreases when the transcript is 37 nt long and IH2 becomes thermodynamically favorable (Supplementary Fig. 19g). Similarly, the reactivity of A14 increases in coordination with decreased reactivity at G19 (IH2 bulge)

and U22 (IH2 loop) and increased reactivity at A33 (L2) when the transcript is 45 nt long, which correlates with P2c and P2b folding and disrupting IH2 so that IH1 can fold again (Supplementary Fig. 19e, g). The reactivity of A14 and U11 then decreases when the transcript is 54 and 56 nts long, respectively, as each nucleotide becomes paired in P2a (Supplementary Fig. 19f, g).

Similar folding transitions were detected in cotranscriptional BzCN and DMS probing data, however, the intermediate structures that occur before P2 folding were less clearly delineated and appear to include a third intermediate hairpin (IH3) that comprises base pairs between nucleotides 24–30 and 36–42 with an internal bulge (Fig. 6b). IH3 is mutually exclusive with IH2 and was not detectable when transcripts were equilibrated before BzCN probing. Formation of P2c was detected as increased BzCN reactivity at A33, which is located in L2, from transcripts ~54 to ~67 in coordination with decreased DMS reactivity at C29 and A35, which pair with G34 and U28 in P2c, respectively. (Fig. 6d, f). P2b folding is coordinated with P2c folding and is detected as decreased BzCN reactivity at G19 and U22, which are not paired in IH1, IH2, or IH3 but become constrained in P2b when G19 pairs with C45 and U22 stacks with G21 and G37[47,48] (Fig. 6a-d, g). The DMS reactivity of C24 and C25, which are unpaired in IH2 and paired in P2b, decreased across the same transcript lengths (Fig. 6a-d, g). In coordination with P2b folding, the reactivity of A14, which is unpaired in IH1 and paired in IH2, increased (Fig. 6c, d, h). Similarly, the BzCN reactivity of U11 in the IH1 loop and the DMS reactivity of C16, which is part of a 3 bp stem in IH1 and a 6 bp stem in IH2, increased to a lesser degree (Fig. 6c, d, h). Because P2b is mutually exclusive with IH2 and reduces the extent to which IH1 can interact with unpaired nucleotides in the nascent transcript, increased IH1 loop reactivity upon P2b folding is likely caused by the collapse of a heterogenous pool of structures into a single folding pathway. These early folding intermediates likely include IH1, IH2, and IH3, but may also include other non-native structures that were not clearly identified in equilibrium chemical probing experiments (Fig. 6b, c). P2a folding is then observed as decreased reactivity at U11, A14, and C16 from transcript ~67 to ~76 as these nucleotides become paired in P2a, and increased DMS reactivity at A18, which forms a Hoogsteen/sugar edge pair with G46 in P2b[47,48] (Fig. 6e, h).

We assessed whether intermediate structures like IH1, IH2, and IH3 are present in other ppGpp riboswitches by predicting minimum free energy structures using the RNAstructure[49] Fold command. 18 of the 30 putative ppGpp aptamers that were assessed were predicted to form an IH1-like structure in which a conserved YCU trinucleotide base pairs with conserved purines at the 5′ end of the aptamer (Supplementary Fig. 20a-c, f). Of these structures, 80% have a minimum free energy that is less than or equal to that of the experimentally verified *Cba* IH1 structure (Supplementary Fig. 20e). IH1 is therefore a consequence of two closely positioned ppGpp aptamer sequence elements having a high likelihood of forming contiguous base pairs. All ppGpp aptamers are predicted to form some non-native structure when the transcript is sufficiently long to form the IH2 and IH3 hairpins observed in the *Cba* ppGpp aptamer (Supplementary Fig. 20g). However, these predicted structures are heterogenous, which is expected because they are composed of base pairs between conserved and variable regions of the aptamer (Supplementary Fig. 20c, d, h).

### Folding of the P3 and P4 hairpins
In principle, P3 can fold once it has fully emerged from RNAP at transcript ~82. With the exception of A64, nucleotides within P3 are not reactive to BzCN and nucleotides U60 and A63 within the L3 are BzCN-reactive, suggesting that P3 does fold (Supplementary Fig. 18a, b). In contrast, nucleotides A58 and C66 within P3 are consistently DMS reactive until after the full aptamer has emerged from RNAP at transcript ~118 (Supplementary Fig. 18c, e). One possible explanation for this apparent contradiction is that P3 may not be stably folded until the entire ppGpp aptamer has folded, and the longer reaction time used

for DMS probing was able to detect the unfolded P3 stem while BzCN probing was not. Notably, ppGpp binding decreased the reactivity of P3 nucleotides, indicating that P3 is stabilized in the context of the full, ligand-bound aptamer (Supplementary Fig. 18c, e). Regardless of when P3 is stably folded, P4 folding was detected as decreased BzCN reactivity at U72 and U73 from transcript 100 to 105, as the downstream P4 stem emerges from RNAP (Supplementary Fig. 18c, e).

### ppGpp binding stabilizes a preformed ligand binding pocket
ppGpp binding was detected at transcript ~123 as decreased DMS reactivity in nucleotides proximal to the ppGpp-binding pocket (Fig. 7a, b). The most direct signature of ppGpp binding is decreased DMS reactivity at C69, which forms a Watson-Crick pair with the G nucleobase of ppGpp[47,48] (Fig. 7b, c). The ppGpp-dependent formation the G10-A68 pair, which stacks with the C69-ppGpp pair[47,48], was detected as decreased DMS reactivity at A68 (Fig. 7b, c). Similarly, the ppGpp-dependent formation of contacts between G48 and A70, which stacks on the opposite face of the C69-ppGpp pair[47,48], was detected as decreased DMS reactivity at A70 (Fig. 7b, c). The DMS reactivity of C49, which stacks with G48 and coordinates a $Mg^{2+}$ that contacts the 3′ terminal phosphate of ppGpp[47,48], and the BzCN reactivity G98, which directly contacts the 5′ terminal phosphate of ppGpp[47,48], also decrease upon ppGpp binding (Fig. 7b, c and Supplementary Fig. 18d, e). In contrast to the ppGpp-dependent changes in the DMS reactivity of nucleotides 68–70, the BzCN reactivity of nucleotides 68–70 decreases at transcript ~115 independent of ppGpp binding (Supplementary Fig. 18d, e). Given that BzCN measures backbone flexibility and DMS measures Watson-Crick face accessibility, this suggests that folding of the apo aptamer partially organizes the ligand binding pocket, but that ppGpp binding is required to stably establish the G10-A68 and G48-A70 pairs. The DMS reactivity of nucleotides A94 and C95 within J4-1, which are proximal to the ppGpp binding pocket[47,48], also decreases upon ppGpp binding (Fig. 7b, c). All ppGpp-dependent reactivity changes described above and in the sections below were abolished by the C69A point mutation, which disrupts ppGpp binding[23] (Fig. 7c and Supplementary Fig. 21).

### ppGpp binding stabilizes a distal tertiary interaction
Crystal structures of ppGpp aptamers revealed that a stretch of purines in P4 forms long-range interactions with the minor groove of P2b[47,48] (Fig. 7a, d). Independent of ppGpp binding, the DMS reactivity of A18, which is the only DMS-reactive (A or C) nucleotide in P2b with an unpaired Watson-Crick face[47,48], and P4 decreases at transcript ~108 (Fig. 7e). This is consistent with the formation of long-range P2b-P4 contacts when P4 has fully emerged from RNAP. In coordination with the signatures of ppGpp binding described above, these P2b and P4 nucleotides undergo a further ppGpp-dependent decrease in DMS reactivity beginning at transcript ~115 as the ppGpp aptamer emerges from RNAP (Fig. 7e). It is important to note that the reactivity changes observed in P4 cannot confidently be attributed to specific nucleotides because P4 contains 6 consecutive A nucleotides, which prevents precise cDNA mutation mapping[20]. Nonetheless, the observation of sequential ppGpp-independent and ppGpp-dependent decreases in the DMS reactivity of P2b and P4 suggests that ppGpp binding stabilizes the interaction between the two helices despite the distance of the P2b-P4 contacts from the ligand binding pocket. Consistent with this interpretation, the ppGpp binding-deficient C69A mutant undergoes the ppGpp-independent decrease in DMS reactivity at transcript ~108 but not the ppGpp-dependent decrease in DMS reactivity at transcript ~115 (Fig. 7e).

### ppGpp binding stabilizes P1 base pairs against terminator hairpin folding
It has been proposed that ppGpp binding promotes transcription antitermination by stabilizing P1, which blocks the propagation of

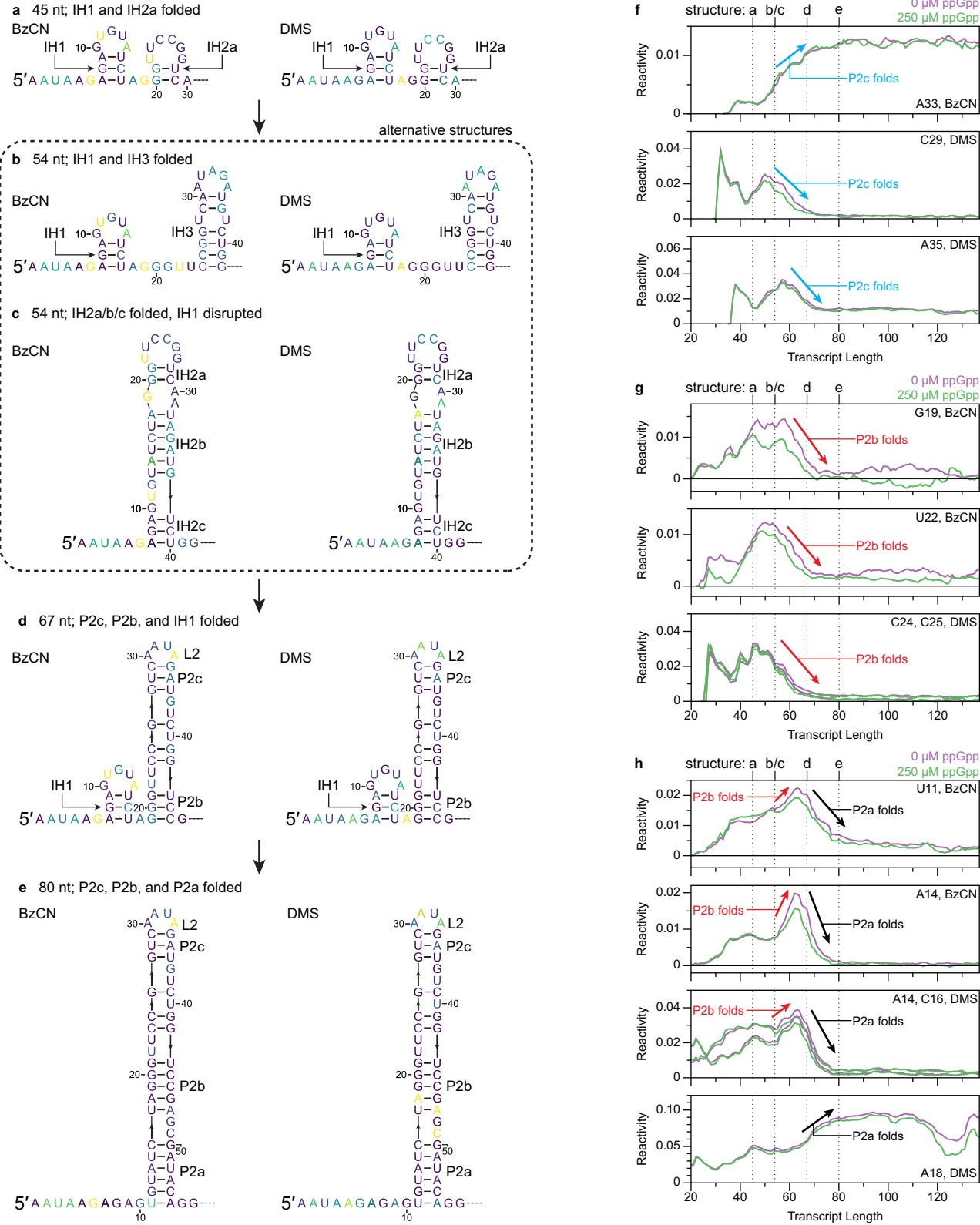

**Fig. 6 | Early folding intermediates of the *C. bacterium* ppGpp riboswitch.**
**a**–**e** Secondary structures of *Cba* ppGpp riboswitch folding intermediates containing IH1 and IH2a (**a**), IH1 and IH3 (**b**), IH2a/b/c (**c**), IH1 and P2b/c (**d**), and P2a/b/c (**e**) colored by the BzCN or DMS reactivity of the indicated transcript in the absence of ppGpp. **f, g, h** Plots showing transcript length-dependent BzCN reactivity changes that occur during P2c (**f**), P2b (**g**), and P2a (**h**) folding. Data from 0 µM and 250 µM ppGpp samples are purple and green, respectively. BzCN reactivity is from Supplementary Fig. 18a, b. DMS reactivity is from Fig. 5d, e. Arrows indicate reactivity changes associated with P2a (black), P2b (red), and P2c (blue) folding. Reactivity is shown as background-subtracted mutation rate. Source data are provided as a Source Data file. BzCN benzoyl cyanide; DMS dimethyl sulfate.

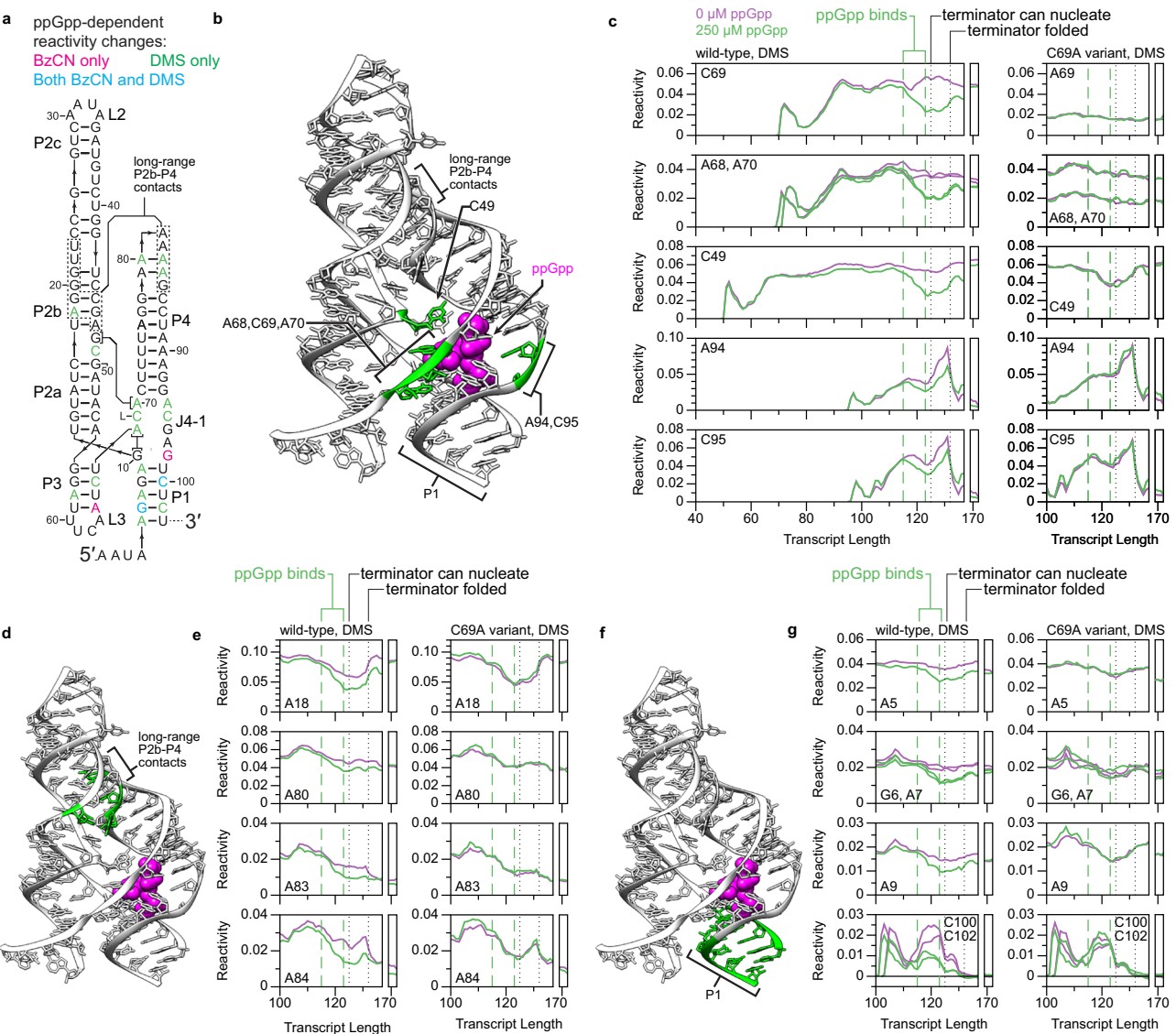

**Fig. 7 | Ligand-dependent changes in ppGpp aptamer structure. a** *Cba* ppGpp aptamer secondary structure colored to show nucleotides that undergo ppGpp-dependent reactivity changes observed using BzCN, DMS, or both probes. **b, d, f** *Thermoanaerobacter mathranii* PRPP riboswitch G96A (ppGpp-binding) variant structure (PDB 6CK4)[47] highlighted to indicate ppGpp-responsive nucleotides in the ligand binding pocket (**b**) and within P2b-P4 contacts (**d**), and ppGpp-responsive base pairs in P1 (**f**). **c, e, g** Plots of transcript length-dependent reactivity changes for ppGpp-responsive nucleotides in the ligand binding pocket (**c**), within P2b-P4 contacts (**e**), and within P1 (**g**) for the WT *Cba* ppGpp riboswitch and C69A variant. Data from 0 μM and 250 μM ppGpp samples are purple and green, respectively. Vertical dotted and dashed lines mark when the indicated folding transitions occur. Data are from Fig. 5d, e and Supplementary Fig. 21a, b. Reactivity is shown as background-subtracted mutation rate. Source data are provided as a Source Data file. DMS dimethyl sulfate.

---

terminator base pairs[47,48]. In agreement with this model, we observed a ppGpp-dependent decrease in the reactivity of every P1 base pair (Fig. 7a, f, g and Supplementary Fig. 18d). The folded ppGpp riboswitch terminator hairpin was detected in cotranscriptional and equilibrium BzCN probing experiments as a low reactivity region spanning nucleotides 76 to 134 (Supplementary Fig. 22a, b). Within this region, A107, which corresponds to the 'A' within the apical GNRA tetraloop, A89, which is predicted to be a single nucleotide bulge, and G78, which is part of a terminator helix extension in which a purine tract pairs with the poly-U tract, were BzCN reactive (Supplementary Fig. 22a, b, e). Similarly, the apical terminator loop was DMS reactive (Supplementary Fig. 22c). In the absence of ppGpp, terminator nucleation was detected as decreased DMS reactivity at C100 and C102, which form base pairs near the apical loop of the terminator, at transcript ~125 when ~4 terminator base pairs can form outside of RNAP (Fig. 7g). In the presence of ppGpp, terminator nucleation at transcript ~125 was coordinated with weakly increased BzCN reactivity in P2b and P4 even though all

observed ppGpp-dependent differences in DMS reactivity were maintained (Fig. 7 and Supplementary Figs. 18b and 22d). It is unclear whether this change in nucleotide flexibility is due to a sub-population of ppGpp aptamers being disrupted by terminator nucleation, or if the emergence of the terminator loop and downstream stem nucleotides from RNAP perturbs the ppGpp aptamer without disrupting it. However, the DMS reactivity of nucleotides that nucleate terminator hairpin folding is higher in the presence of ppGpp, suggesting that at least some ppGpp aptamers resist terminator nucleation (Supplementary Fig. 22f). Similarly, the DMS reactivity of A89, which is paired in the ppGpp aptamer and unpaired in the terminator, is lower in the presence of ppGpp after the terminator hairpin can completely fold at transcript +133 (Supplementary Fig. 22f). When RNAP reaches the primary termination sites at positions +133/134, completion of the terminator stem is observed as increased BzCN reactivity at the A89 bulge and a partial reduction of ppGpp-dependent differences in DMS reactivity as some ppGpp-bound aptamers are disrupted by the

terminator hairpin (Fig. 7 and Supplementary Fig. 22e). In the ZTP and fluoride riboswitches described above, ligand-dependent structures persisted after RNAP had bypassed the transcription termination site. In contrast, ppGpp-dependent reactivity signatures were not detectable in transcripts that accumulated at the terminal transcription roadblock downstream of the termination site (Fig. 7 and Supplementary Fig. 22f, transcripts 169 to 172). One possible explanation for this effect is that the dissociation of ppGpp from the aptamer allows the terminator to nucleate. This is likely possible because the downstream terminator hairpin stem is predicted to form a weak hairpin that does not block the formation of the U103-A108 nucleating base pair (Supplementary Fig. 22g).

## Discussion

We have demonstrated that TECprobe-VL detects biologically meaningful RNA folding intermediates at nucleotide resolution and delineates when coordinated structural rearrangements can occur within a cotranscriptional RNA folding pathway. TECprobe-VL simplifies and standardizes cotranscriptional RNA structure probing experiments by using MaP[20] to encode RNA structural information in cDNA, and by structuring cDNA as an amplicon using a 5′ structure cassette. This approach eliminates the need for size selection procedures that can cause information loss near transcript 3′ ends, ensures even sequencing coverage across the length of each transcript, and enables the possibility of integrating TECprobe-VL with MaP-based methods for resolving structural heterogeneity[50–54] and detecting RNA-protein interactions[55]. The benefits of this approach are evident in our reinvestigation of ZTP and fluoride riboswitch folding and our analysis of ppGpp riboswitch folding: in all cases, improved resolution of 3′-proximal RNA structures enabled the detection of cotranscriptional folding events in which coordinated reactivity changes spanned the entire transcript. In combination with complementary biophysical approaches[8–13], the ability to resolve coordinated structural changes by cotranscriptional RNA chemical probing will likely aid efforts to predict how RNA folds cotranscriptionally[56–62].

There are two potential drawbacks to structuring cDNA as an amplicon in the TECprobe-VL procedure: first, structuring cDNA as an amplicon requires the inclusion of a 5′ structure cassette in the target RNA sequence, which could potentially influence RNA folding in some cases. The use of structure cassettes in RNA chemical probing experiments is well-validated[63] and the SC1 structure cassette did not interfere with the function or folding of the three riboswitches that were investigated in this study. However, two alternative structure cassettes that were less compact than SC1 did perturb ZTP riboswitch function (Supplementary Fig. 23). In cases where a 5′ structure cassette cannot be appended to the target RNA, it may be possible to use a hybridization-based strategy to append an adapter to the cDNA 3′ end[26,28]. The main disadvantage of this approach in cotranscriptional RNA structure probing experiments is that it is currently difficult to remove adapter dimer without also depleting short cDNAs. Second, obtaining complete coverage of cDNA requires the use of sequencing reads that span the length of the longest intermediate transcript in the library. Consequently, the upper bound for target RNA length is currently ~440 nt, which accounts for a constant primer binding site downstream of the target RNA sequence, the 5′ structure cassette, barcoding modified and untreated samples, and a buffer to accommodate insertions in the cDNA that can occur during error-prone reverse transcription. At the time of writing, the maximum target RNA length we have tested is 273 nt. These limitations are offset by the advantages of using a 5′ structure cassette to circumvent the need to ligate an adapter to the cDNA 3′ end. Most importantly, this eliminates the need to deplete adapter dimer, which caused poor resolution of 3′-proximal structures in the original cotranscriptional SHAPE-seq procedure due to the loss of library fragments with short inserts during size selection. In the reanalysis of the ZTP and fluoride riboswitches by

TECprobe-VL, resolving 3′-proximal RNA structures enabled the detection of previously unobservable folding events and uncovered additional evidence in support of known folding events. In the *pfl* ZTP riboswitch system, it became possible to detect the direct interaction of ZMP with U94 and G95 in L3 and the ZMP-dependent stabilization of the G84-C98 base pair, which must be broken for the terminator hairpin to nucleate (Supplementary Fig. 10c, d). In addition, the IH1-to-P1 folding transition was detected as both decreased IH1 loop reactivity (previously observed) and decreased downstream P1 stem reactivity (not previously observed) (Supplementary Fig. 9a, b). Similarly, in the fluoride riboswitch system, pseudoknot folding was detected as decreased reactivity in both the upstream (previously observed) and downstream (not previously observed) segments of the pseudoknot (Supplementary Fig. 16a, b). The ability to detect cotranscriptional folding events as coordinated reactivity changes that span the length of a transcript is likely to aid both manual and automated[64] interpretation of cotranscriptional RNA structure probing data.

The collection of BzCN and DMS data sets for three riboswitch systems enabled us to compare the performance of each probe in cotranscriptional RNA chemical probing experiments. Whereas the BzCN probing reaction was performed using single-hit conditions and self-quenches in ~1 s, the DMS probing reaction was performed using multiple-hit conditions and was quenched after a 5 min reaction. Despite the longer reaction time for DMS probing, the same folding intermediates were observed with both probes for all riboswitch targets. However, it is important to note that transcription was allowed to proceed for 2 min before BzCN or DMS probing was started. As described further below, this could allow cotranscriptionally folded RNA that is displayed from RNAP equilibrate to some degree before the chemical probing reaction begins. This limitation cannot be circumvented because it is essential to let the single-round transcription reaction run to completion before chemical probing begins. One possible consequence of probing with DMS for several minutes is that static transcription complexes can potentially backtrack over time, which displaces the RNA 3′ end from the RNAP active center. Because the location of RNAP is inferred from the RNA 3′ end, backtracking can shift the apparent transcript length at which a folding transition occurs forward[15]. While most folding transitions occurred at the same transcript length in both BzCN and DMS data sets, the decrease in reactivity of the IH1 loop and downstream P1 stem of the ZTP aptamer during the IH1-to-P1 folding transition reached its lowest point at transcript 80 with BzCN and transcript ~86 with DMS (Fig. 3e and Supplementary Fig. 8c). In several cases BzCN and DMS probing provided complementary information. For example, in the *pfl* ZTP riboswitch the stabilization of a Hoogsteen/Hoogsteen base pair between G19 and A47 was detected as simultaneously decreased BzCN reactivity and increased DMS reactivity at A47 upon ZMP binding (Supplementary Fig. 11f, h). Similarly, in the *crcB* fluoride riboswitch the fluoride-dependent formation of a Hoogsteen base pair between A40 and U48 was detected as simultaneously decreased BzCN reactivity and increased DMS reactivity at A40 (Fig. 4f). We therefore suggest that it is useful to perform parallel cotranscriptional RNA structure probing experiments with both BzCN and DMS when possible. DMS probing will likely be advantageous when characterizing long RNA targets due its superior signal-to-noise relative to BzCN probing and when identifying sub-populations in heterogenous mixtures[50–54]. In addition, because in vitro transcription is typically performed at pH 8.0 (as it is in the current TECprobe procedures), cotranscriptional RNA structure probing experiments are likely to be compatible with four-base DMS probing[65].

The usefulness of parallel BzCN and DMS probing data sets was most evident in the detection of ppGpp binding by the *Cba* ppGpp riboswitch. In contrast to the *pfl* ZTP and *crcB* fluoride riboswitches, little evidence of ppGpp-dependent aptamer stabilization was detected by BzCN probing (Supplementary Fig. 18). Upon complete aptamer

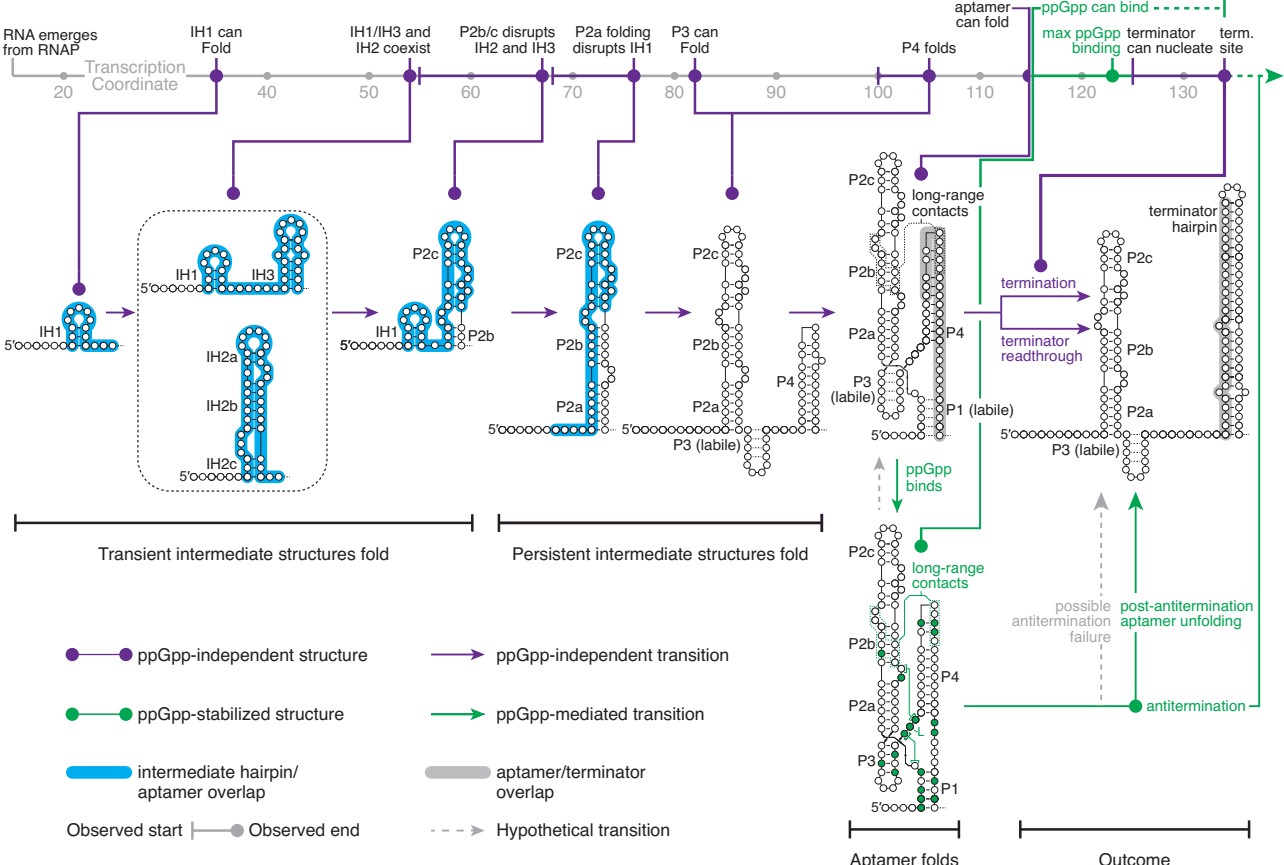

**Fig. 8 | A model for *Cba* ppGpp riboswitch folding.** *Cba* riboswitch folding intermediates as observed by TECprobe-VL are plotted by transcription coordinate. Purple indicates ppGpp-independent structures and transitions; Green indicates ppGpp-stabilized structures or ppGpp-mediated transitions; gray indicates hypothetical transitions. Initial *Cba* ppGpp aptamer folding comprises the formation of mutually exclusive transient intermediates (IH1/IH3 and IH2) that are disrupted upon P2 folding. P3 and P4 are then able to fold once the nascent transcript is sufficiently long. However, P3 may not be stably folded outside the context of the

ppGpp-bound aptamer. Once the downstream P1 stem has emerged from RNAP, the aptamer becomes competent to bind ppGpp. P1 is labile in the absence of ppGpp, which allows the terminator hairpin to fold and terminate transcription. ppGpp stabilizes numerous long-range contacts within the aptamer, including P1 base pairs, which inhibits terminator folding and allows RNAP to bypass the termination site. Eventually, the terminator hairpin is able to fold, either by disrupting the ppGpp-bound aptamer or by disrupting the apo aptamer after ppGpp dissociates.

folding, the BzCN reactivity of nucleotides within the ligand binding pocket decreased regardless of whether ppGpp was present. This indicates that the flexibility of nucleotides within the ligand binding pocket is approximately equivalent regardless of whether ppGpp is bound and that the ligand binding pocket is at least partially organized. However, DMS probing revealed that the G10-A68 pair and contacts between G48 and A70 do not stably form until C69 pairs with the G nucleobase of ppGpp. Together, these contacts directly stabilize P1 by extending its helical stack[47,48]. DMS probing also revealed that ppGpp binding indirectly stabilizes distal contacts between P2b and P4 and several base pairs in P3. These ligand-dependent changes in structure presumably stabilize P1 indirectly by maintaining the four-way junction structure of the folded ppGpp aptamer. Taken together, these observations indicate that the apo ppGpp aptamer is folded into a four-way junction structure and suggest that ppGpp binding stabilizes P1 directly by extending the P1 stack and indirectly by stabilizing long-range contacts between P2 and P4, thereby conferring terminator resistance.

Analysis of ppGpp riboswitch folding using TECprobe-VL established a positional map of folding intermediates that can occur as the *Cba* ppGpp aptamer folds (Fig. 8). Like the *pfl* ZTP aptamer, transient intermediate hairpins precede *Cba* ppGpp aptamer folding. The first transient structure, IH1, is potentiated by the high likelihood of forming contiguous base pairs between conserved poly-purine and

poly-pyrimidine stretches within the ppGpp aptamer and therefore may be present in other ppGpp riboswitch folding pathways. Minimum free energy structure predictions suggest that the subsequent transient structures that were observed for the *Cba* ppGpp riboswitch, IH2 and IH3, are not present in other ppGpp riboswitch folding pathways (Supplementary Fig. 20). However, all putative ppGpp riboswitches that were assessed are predicted to form non-native structures that precede P2 hairpin folding. Determining whether the observed transient structures function as 'folding guides'[66] that coordinate *Cba* ppGpp aptamer folding is not possible using the current data and will likely require approaches for evaluating how perturbing these structures affects riboswitch function. In addition to the *pfl* ZTP and *Cba* ppGpp riboswitches, transient non-native structures have also been detected within the folding pathways of the *E. coli* signal recognition particle, RNase P, tmRNA, and *thiB* riboswitch RNAs[14,16,67,68]. In most of these cases, the non-native structure precedes the folding of a native structure that requires base pairs between distal nucleotides. While some transient structures are conserved[56,61], others, such as the *pfl* ZTP riboswitch IH1 hairpin and *Cba* ppGpp riboswitch IH2 and IH3 hairpins, appear to be a consequence of the nascent transcript sampling energetically favorable structures until it is long enough that the native structure can fold. The occurrence of non-native structures in such contexts is not surprising given the capacity of random RNA sequences for the formation of secondary structures[69,70]. In such cases,

intermediate structures that can be easily resolved to permit the native structure to fold are likely advantageous. This is likely to be particularly important for riboswitches that regulate transcription because the resolution of non-native structures must occur cotranscriptionally.

Our validation and benchmarking of TECprobe-VL focused on comparisons with Cotranscriptional SHAPE-Seq because the two methods use closely related in vitro transcription procedures in which RNAP is halted at biotin-streptavidin roadblocks under single-round conditions. In vitro cotranscriptional RNA structure probing has also been performed using an orthogonal procedure, called Structural Probing of Elongating Transcripts (SPET-seq)[16]. In an in vitro SPET-seq experiment, a multi-round in vitro transcription reaction is halted by the simultaneous addition of actinomycin D and DNaseI before nascent RNA is chemically probed. The resulting intermediate transcripts are then further enriched by gel extraction before conversion into a sequencing library. We prefer to use template DNA strand biotin-streptavidin complexes to distribute RNA polymerase across the DNA template because they halt transcription with >80% efficiency and the resulting complexes are >99% stable[15]. This enables the use of single-round in vitro transcription, which simplifies the transcription reaction. However, in cases where multi-round transcription conditions are advantageous or preferred, all of the sequencing library construction optimizations that we have described are generalizable to SPET-seq. Furthermore, the use of neighboring transcript smoothing to address uneven intermediate transcript representation is generalizable to all cotranscriptional RNA structure probing methods.

Like all cotranscriptional RNA structure probing methods in which nascent RNA is chemically probed in the context of a road-blocked RNAP, TECprobe-VL has two primary limitations. First, TECprobe-VL does not measure true cotranscriptional folding because transcription must be artificially halted at a roadblock before chemical probing can be performed. Given that base pair formation occurs on a microsecond timescale[4], halting transcription could permit local equilibrium of the cotranscriptionally folded RNA. Second, although it is useful to visualize reactivity data from cotranscriptional RNA structure probing experiments as a continuous trajectory as we have done above, the profile of each intermediate transcript is technically an end-point measurement. Therefore, current experiments cannot definitively link the intermediate structures that are observed within a cotranscriptional folding pathway. Third, the current method uses *E. coli* RNAP to transcribe RNA regardless of the origin of the target RNA. This could potentially influence RNA folding outcomes if the transcription pausing properties or RNAP-nascent RNA interactions of *E. coli* RNAP are different than those of the cognate RNAP[3–5]. Notably, the first two of these limitations are complementary to those of biophysical approaches, which have high temporal resolution and track RNA folding continuously, but which do not resolve RNA secondary or tertiary structure directly. Furthermore, the high efficiency of TECprobe-VL will likely enable its use as the basis for developing new cotranscriptional RNA structure probing procedures that overcome the limitations of current method.

## Methods
The source and catalog number (if applicable) for all reagents are provided as a Supplementary Data 1.

### Oligonucleotides
All oligonucleotides were purchased from Integrated DNA Technologies. A detailed description of all oligonucleotides including sequence, modifications, and purifications is presented in Supplementary Table 1.

### Proteins
Q5 High-Fidelity DNA Polymerase, Vent (exo-) DNA polymerase, *Sulfolobus* DNA Polymerase IV, *E. coli* RNA Polymerase holoenzyme, Mth RNA Ligase (as part of the 5′ DNA Adenylation kit), T4 RNA Ligase 2

truncated KQ, ET SSB, RNase H, and RNase I$_f$ were purchased from New England Biolabs. TURBO DNase, SuperaseIN, SuperScript II, SuperScript III, and BSA were purchased from ThermoFisher. Streptavidin was purchased from Promega. NusA was a gift from J. Roberts (Cornell University).

### DNA template purification
Supplementary Table 2 provides details for the oligonucleotides and processing steps used for every DNA template preparation in this work. Supplementary Table 3 provides DNA template sequences.

5′ biotinylated DNA that was used for transcription anti-termination assays and as a PCR template when preparing modified DNA templates below was PCR amplified from plasmid DNA using Q5 High-Fidelity DNA Polymerase (New England Biolabs) and primers HP4_5bio.R and PRA1_NoMod.F (for ZTP and fluoride templates) or PRA1_shrt.F (for ppGpp templates) (Supplementary Table 1). PCR was performed as three 100 µl reactions containing 1X Q5 Buffer (New England Biolabs), 1X Q5 High GC Enhancer (New England Biolabs), 0.2 mM dNTP Solution Mix (New England Biolabs), 250 nM PRA1_NoMod.F or PRA1_shrt.F, 250 nM HP4_5bio.R, 20 pM template DNA, and 0.02 U/µl Q5 DNA polymerase using the following thermal cycler protocol with a heated lid set to 105 °C: 98 °C for 30 s, [98 °C for 10 s, 65 °C for 20 s, 72 °C for 20 s] x 30 cycles, 72 °C for 5 min, hold at 10 °C. The resulting PCR product was purified by UV-free agarose gel extraction using a QIAquick gel extraction kit[71]. A step-by-step protocol for this procedure is available[72].

Randomly biotinylated DNA templates for TECprobe-VL experiments were PCR amplified from a 5′ biotinylated linear DNA template using Vent (exo-) DNA polymerase (New England Biolabs) and primers HP4_5bio.R and PRA1_NoMod.F (for ZTP and fluoride templates) or PRA1_shrt.F (for ppGpp templates) (Supplementary Table 1). 200 µl PCRs contained 1X ThermoPol Buffer (New England Biolabs), 250 nM PRA1_NoMod.F or PRA1_shrt.F, 250 nM HP4_5bio.R, 20 pM template DNA, 0.02 U/µl Vent (exo-) DNA polymerase, 200 µM dNTP Solution Mix, and a concentration of biotin-11-dNTPs (PerkinElmer, Biotium) that favored the incorporation of ~2 biotin modifications[15] in the transcribed region of each DNA template. PCR was performed as two 100 µl reactions in thin-walled tubes using the following thermal cycler protocol with a heated lid set to 105 °C: 95 °C for 3 min, [95 °C for 20 s, 58 °C for 30 s, 72 °C for 30 s] x 30 cycles, 72 °C for 5 min, hold at 12 °C. PCR products were purified as described below in the section *SPRI bead purification of DNA*, eluted into 50 µl of 10 mM Tris-HCl (pH 8.0), and quantified using the Qubit dsDNA Broad Range Assay Kit (Invitrogen) with a Qubit 4 Fluorometer (Invitrogen). A step-by-step protocol for this procedure is available[72].

DNA templates for TECprobe-SL experiments, which contained an internal etheno-dA stall site, were prepared using a variation of a previously described[24,71] procedure for preparing DNA that contains a chemically-encoded transcription stall site. In this variation of the procedure, PCR products were purified using SPRI beads instead of by column purification. DNA with an internal etheno-dA stall site was PCR amplified from an unmodified linear DNA template using primers PRA1_NoMod.F and dRP1iEthDA.R[24,71] (Supplementary Table 1). PCR was performed as three 100 µl reactions containing 1X Q5 Buffer, 1X Q5 High GC Enhancer, 0.2 mM dNTPs, 250 nM PRA1_NoMod.F, 250 nM dRP1iEthDA.R, 20 pM template DNA, and 0.02 U/µl Q5 DNA polymerase using the following thermal cycler protocol with a heated lid set to 105 °C: 98 °C for 30 s, [98 °C for 10 s, 65 °C for 20 s, 72 °C for 20 s] x 30 cycles, 72 °C for 5 min, hold at 10 °C. PCR products were purified as described below in the section *SPRI bead purification of DNA* and eluted into 100 µl of 10 mM Tris-HCl (pH 8.0) per 300 µl PCR. Translesion DNA synthesis using *Sulfolobus* DNA polymerase IV (New England Biolabs) was performed by incubating two 100 µl reactions containing the purified PCR product, 1X ThermoPol Buffer, 0.2 mM dNTPs, 0.02 U/µl Vent (exo-) DNA Polymerase, and 0.02 U/µl

*Sulfolobus* DNA Polymerase IV at 55 °C for one hour in a thermal cycler with a heated lid set to 105 °C[24,71]. After translesion DNA synthesis, DNA templates were purified as described below in the section *SPRI bead purification of DNA*, eluted into 75 µl of 10 mM Tris-HCl (pH 8.0), and quantified using the Qubit dsDNA Broad Range Assay Kit with a Qubit 4 Fluorometer. DNA templates for experiments that evaluated reverse transcription additives were prepared as above except dRP1iBio.R, which contains an internal biotin-TEG modification, was used.

## SPRI bead purification of DNA

SPRI beads were prepared in-house using the 'DNA Buffer' variation of the procedure by Jolivet and Foley[73]. Samples were mixed with an equal volume of SPRI beads, incubated at room temperature for 5 min, and placed on a magnetic stand for 3 min so that the beads collected on the tube wall. The supernatant was aspirated and discarded, and the beads were washed twice by adding a volume of 70% ethanol at least 200 µl greater than the combined volume of the sample and SPRI beads to the tube without disturbing the bead pellet while it remained on the magnetic stand. The samples were incubated at room temperature for 1 min before aspirating and discarding the supernatant. Residual ethanol was evaporated by placing the open microcentrifuge tube in a 37 °C dry bath for ~15 s with care taken to ensure that the beads did not dry out. Purified DNA templates were eluted by resuspending the beads in a variable amount of 10 mM Tris-HCl (pH 8.0) (depending on the procedure, details are in each relevant section), allowing the samples to sit undisturbed for 3 min, placing the sample on a magnetic stand for 1 min so that the beads collected on the tube wall, and transferring the supernatant, which contained purified DNA, into a screw-cap tube with an O-ring.

## Transcription antitermination assays

Single-round in vitro transcription using terminally biotinylated template DNA was performed as described for TECprobe-VL below in the section *Single-round* in vitro *transcription for TECprobe-VL and TECprobe-SL* except that the reaction volume was 25 µl for ZTP and ppGpp templates or 50 µl for fluoride templates, 100 nM streptavidin (Promega) was included in the master mix, and the transcription reactions were stopped by adding 75 µl (for ZTP and ppGpp templates) or 150 µl (for fluoride templates) of TRIzol LS without performing RNA chemical probing. RNA was purified as described in the section *RNA purification*, except that volumes of chloroform and isopropanol were scaled to account for the doubled reaction volume for fluoride templates and all samples were dissolved in 6 (ppGpp samples) or 15 µl (ZTP and fluoride samples) of formamide loading dye (90% (v/v) deionized formamide, 1X transcription buffer (defined below), 0.05% (w/v) bromophenol blue, 0.05% (w/v) xylene cyanol FF), and subsequently analyzed as described in the section *Denaturing Urea-PAGE*.

## Denaturing urea-PAGE

Samples in formamide loading dye were heated at 95 °C for 5 min, and snap-cooled on ice for 2 min. Denaturing Urea-PAGE was performed using 8 or 10% gels prepared using the SequaGel UreaGel 19:1 Denaturing Gel System (National Diagnostics) for a Mini-PROTEAN Tetra Vertical Electrophoresis Cell by filling the outer buffer chamber so that it covered ~1 cm of the gel plates, pre-running the gel at 480 V for 30 min, loading the denatured samples, and running the gel at 480 V[74]. Gels were stained with 1X SYBR Gold Nucleic Acid Stain (Invitrogen) in 1X TBE for 10 min and scanned on an Azure Biosystems Sapphire Biomolecular Imager using the 488 nM/518BP22 setting. Uncropped source gels are available in Supplementary Fig. 24.

## TECprobe-VL and TECprobe-SL

We previously described the theory and best practices for cotranscriptional RNA structure probing experiments, and detailed procedures for randomly biotinylated DNA template preparation, cotranscriptional RNA chemical probing, RNA purification, and 3' adapter ligation[72].

## Single-round in vitro transcription for TECprobe-VL and TECprobe-SL

All single-round in vitro transcription reactions for TECprobe-VL experiments were performed as 60 µl reactions containing 1X Transcription Buffer [20 mM Tris-HCl (pH 8.0), 50 mM KCl, 1 mM dithiothreitol (DTT), and 0.1 mM EDTA], 0.1 mg/ml Molecular Biology-Grade BSA (Invitrogen), 100 µM high-purity NTPS (Cytiva), 10 nM randomly biotinylated template DNA, and 0.024 U/µl *E. coli* RNA polymerase holoenzyme (New England Biolabs). Single-round in vitro transcription reactions for TECprobe-SL experiments were performed as 60 µl reactions containing the same reagents as TECprobe-VL experiments except that the reactions used a 10 nM template DNA that contains an internal etheno-dA stall site downstream of a *Cbe pfl* ZTP riboswitch variant in which the poly-U tract was removed, and either 100 or 500 µM high-purity NTPs, as indicated. Transcription reactions for experiments where DMS was used to probe RNA structures contained additional Tris-HCl (pH 8.0) at final concentration of 100 mM to minimize pH changes during the chemical probing reaction. At the time of preparation, each TECprobe-VL reaction was 48 µl due to the omission of 10X (1 µM) streptavidin (Promega) and 10X Start Solution [100 mM MgCl₂, 100 µg/ml rifampicin (Gold Biotechnology)] from the reaction. Each TECprobe-SL reaction was 54 µl due to the omission of 10X Start Solution from the reaction.

The composition of the single-round in vitro transcription master mix varied depending on the riboswitch system that was assessed. In vitro transcription reactions for the *Cbe pfl* ZTP riboswitch contained 2% (v/v) DMSO and, when present, 1 mM ZMP (Sigma-Aldrich). In vitro transcription reactions for the *Bce crcB* fluoride riboswitch that were performed in the presence of fluoride contained 10 mM NaF (Sigma-Aldrich). In vitro transcription reactions for the *Cba* ppGpp riboswitch and C69A ppGpp riboswitch variant contained 500 nM NusA and, when present, 250 µM ppGpp (Guanosine-3′,5′-bisdiphosphate) (Jena Bioscience).

Single-round in vitro transcription reactions were incubated at 37 °C for 10 min to form open promoter complexes. For TECprobe-VL reactions, 6 µl of 1 µM streptavidin was then added for a final concentration of 100 nM streptavidin, and reactions were incubated for an additional 10 min at 37 °C; TECprobe-SL reactions did not include streptavidin but were still incubated for a total of 20 min at 37 °C. Transcription was initiated by adding 6 µl of 10X Start Solution to the reaction for a final concentration of 10 mM MgCl₂ and 10 µg/ml rifampicin. The transcription reaction was incubated at 37 °C for 2 min before chemical probing was performed as described below in the section *RNA chemical probing*.

**RNA chemical probing.** Benzoyl cyanide (BzCN) probing was performed by splitting the sample into 25 µl aliquots and mixing with 2.78 µl of 400 mM BzCN (Sigma-Aldrich) dissolved in anhydrous DMSO (Sigma-Aldrich) [(+) sample)] or with anhydrous DMSO [(-) sample][37,38]. Given that the BzCN probing reaction is complete within ~1 s, 75 µl of TRIzol LS reagent (Invitrogen) was added to each sample without any additional incubation time to stop the in vitro transcription reaction and the samples were vortexed.

Dimethyl sulfate (DMS) probing was performed by splitting the sample into 25 µl aliquots and mixing with 2.78 µl of 6.5% (v/v) DMS (Sigma-Aldrich) in anhydrous ethanol (Sigma-Aldrich) [(+) sample)] or with anhydrous ethanol [(-) sample] and incubating the samples at 37 °C for 5 min. The DMS probing reaction was quenched by adding beta-mercaptoethanol to 2.8 M and incubating the sample at 37 °C for 1 min. 75 µl of TRIzol LS reagent was added to each sample to stop the in vitro transcription reaction and the samples were vortexed.

**RNA purification.** Samples, which contained 27.78 µl (BzCN probing) or 34.45µl (DMS probing) of the cotranscriptional RNA chemical probing reaction in 75 µl of TRIzol LS, were extracted as follows: 20 µl of chloroform was added to each sample, and the samples were mixed by vortexing and inverting the tube and centrifuged at 18,500 x g and 4 °C for 5 min. The aqueous phase was transferred to a new tube and precipitated by adding 1.5 µl of GlycoBlue Coprecipitant (Invitrogen) and 50 µl of ice-cold isopropanol and incubating at room temperature for 15 min. The samples were centrifuged at 18,500 x g and 4 °C for 15 min, the supernatant was aspirated and discarded, 500 µl of ice cold 70% ethanol was added to each sample, and the tubes were gently inverted to wash the samples. The samples were centrifuged at 18,500 x g and 4 °C for 2 min and the supernatant was aspirated and discarded. The samples were centrifuged again briefly to pull down residual liquid, which was aspirated and discarded. The pellet was then resuspended in 25 µl of 1X TURBO DNase buffer (Invitrogen), mixed with 0.75 µl of TURBO DNase (Invitrogen), and incubated at 37 °C for 15 min. 75 µl of TRIzol LS reagent was added to stop the reactions and a second TRIzol extraction was performed as described above, except that the pellet was resuspended in 5 µl of 10% (v/v) DMSO for TECprobe-VL reactions or 25 µl 1X Buffer TM (1X Transcription Buffer, 10 mM MgCl₂) for TECprobe-SL reactions. TECprobe-VL sample processing continued with the *RNA 3′ adapter ligation* section immediately below, while TECprobe-SL samples proceeded directly to the *cDNA synthesis and cleanup* section because the RNA 3′ adapter sequence was included in the transcript.

**RNA 3′ adapter ligation.** 9N_VRA3 adapter oligonucleotide (Supplementary Table 1) was pre-adenylated with the 5′ DNA Adenylation Kit (New England Biolabs) according to the manufacturer's protocol at a 5X scale. Briefly, 100 µl of a master mix that contained 1X DNA Adenylation Buffer (New England Biolabs), 100 µM ATP, 5 µM 9N_VRA3 oligo, and 5 µM Mth RNA Ligase (New England Biolabs) was split into two 50 µl aliquots in thin-walled PCR tubes and incubated at 65 °C in a thermal cycler with a heated lid set to 105 °C for 1 h. Following the reaction, 150 µl of TRIzol LS reagent was added to each 50 µl reaction and the samples were extracted as described above in the section *RNA purification*, except that reaction volumes were scaled to account for the 50 µl reaction volume (40 µl of chloroform was added to the sample-TRIzol mixture and 100 µl of isopropanol was added during the precipitation step). Samples were pooled by resuspending the pellets from each TRIzol extraction in a single 25 µl volume of TE Buffer (10 mM Tris-HCl (pH 7.5), 0.1 mM EDTA). The concentration of the adenylated oligonucleotide was determined using the Qubit ssDNA Assay Kit (Invitrogen) with a Qubit 4 Fluorometer. The molarity of the linker was calculated using 11,142 g/mol as the molecular weight. The adenylation reaction was assumed to be 100% efficient. The linker was diluted to 0.9 µM and aliquoted for future use; aliquots were used within 3 freeze-thaw cycles.

20 µl RNA 3′ adapter ligation reactions were performed by combining purified RNA in 5 µl of 10% DMSO (v/v) from the *RNA purification* section with 15 µl of an RNA ligation master mix such that the final 20 µl reaction contained purified RNA, 2.5% (v/v) DMSO, 1X T4 RNA Ligase Buffer (New England Biolabs), 0.5 U/µl SuperaseIN (Invitrogen), 15% (w/v) PEG 8000, 45 nM 5′-adenylated 9N_VRA3 adapter, and 5 U/µl T4 RNA Ligase 2, truncated, KQ (New England Biolabs). The samples were mixed by pipetting and incubated at 25 °C for 2 h.

**SPRI bead purification of RNA.** Excess 9N_VRA3 3′ adapter oligonucleotide was depleted using a modified SPRI bead purification that contains isopropanol[75]. 17.5 µl of nuclease-free water and 40 µl of freshly aliquoted anhydrous isopropanol (Sigma-Aldrich) were added to each 20 µl RNA ligation reaction, and the samples were mixed by vortexing. Each sample was then mixed with 22.5 µl of SPRI beads so

that the concentration of PEG 8000 was 7.5% (w/v) and the concentration of isopropanol was 40% (v/v) in a sample volume of 100 µl. The samples were incubated at room temperature for 5 min, and placed on a magnetic stand for at least 3 min so that the beads collected on the tube wall. The supernatant was aspirated and discarded, and the beads were washed twice by adding 200 µl of 80% (v/v) ethanol to the tubes without disturbing the bead pellet while it remained on the magnetic stand, incubating the samples at room temperature for 1 min, and aspirating and discarding the supernatant. After discarding the second 80% (v/v) ethanol wash, the sample was briefly spun in a mini centrifuge, and placed back onto a magnetic stand for 1 min to collect the beads on the tube wall. The supernatant was aspirated and discarded, and the beads were briefly (<15 s) dried in a 37 °C dry bath with the cap of the microcentrifuge tube left open. Purified RNA was eluted by resuspending the beads in 20 µl of 10 mM Tris-HCl (pH 8.0), incubating the sample for 3 min at room temperature, placing the sample on a magnetic stand for 1 min so that the beads collected on the tube wall, and transferring the supernatant into a clean microcentrifuge tube. The eluted RNA was mixed with 11.5 µl of RNase-free water and 40 µl of anhydrous isopropanol. 6 µl of 50% (w/v) PEG 8000 and 22.5 µl of SPRI beads were added to each sample so that the concentration of PEG 8000 was 7.5% (w/v) and the concentration of isopropanol was 40% (v/v) in a sample volume of 100 µl. The RNA was purified as described above a second time, except that the RNA was eluted into 25 µl of 1X Buffer TM.

## cDNA synthesis and cleanup

5 µl of 10 mg/ml Dynabeads MyOne Streptavidin C1 beads (Invitrogen) per sample volume were equilibrated in Buffer TX (1X Transcription Buffer, 0.1% (v/v) Triton X-100)[74]. Briefly, after placing the beads on a magnet stand and removing the storage buffer, the beads were resuspended in 500 µl of Hydrolysis Buffer (100 mM NaOH, 50 mM NaCl) and incubated at room temperature for 10 min with rotation. Hydrolysis Buffer was removed, and the beads were resuspended in 1 ml of High Salt Wash Buffer (50 mM Tris-HCl (pH 7.5), 2 M NaCl, 0.5% (v/v) Triton X-100), transferred to a new tube, and washed by rotating for 5 min at room temperature. High Salt Wash Buffer was removed, and the beads were resuspended in 1 ml of Binding Buffer (10 mM Tris-HCl (pH 7.5), 300 mM NaCl, 0.1% (v/v) Triton X-100), transferred to a new tube, and washed by rotating for 5 min at room temperature. After removing Binding Buffer, the beads were washed twice with 500 µl of Buffer TX (1X Transcription Buffer supplemented with 0.1% (v/v) Triton X-100) by resuspending the beads, transferring them to a new tube, washing with rotation for 5 min at room temperature, and removing the supernatant. After washing the second time with Buffer TX, the beads were resuspended to a concentration of ~2 µg/µl in Buffer TX (25 µl per sample volume), split into 25 µl aliquots, and stored on ice until use.

The reverse transcription primer, dRP1_5Bio.R (Supplementary Table 1), contains a 5′ biotin modification and anneals to the 9N_VRA3 adapter (TECprobe-VL) or the transcribed VRA3 sequence downstream of the riboswitch (TECprobe-SL). 1 µl of 500 nM dRP1_5Bio.R was added to the purified RNA, and the samples were incubated on a thermal cycler with a heated lid set to 105 °C using the 'RT anneal' protocol: pre-heat to 70 °C, 70 °C for 5 min, ramp to 50 °C at 0.1 °C/s, 50 °C for 5 min, ramp to 40 °C at 0.1 °C/s, 40 °C for 5 min, cool to 25 °C. Equilibrated Streptavidin C1 beads were placed on a magnetic stand to collect the beads on the tube wall, and the supernatant was removed. The bead pellets were resuspended using the samples (which contained dRP1_5Bio.R oligo annealed to RNA), and incubated at room temperature for 15 min on an end-over-end rotator set to ~8 rpm. The samples were placed on a magnetic stand to collect the beads on the tube wall, the supernatant was aspirated and discarded, and the beads were resuspended in 19.5 µl of reverse transcription master mix, which omits SuperScript II reverse transcriptase at this time. After the 20 µl

reverse transcription reaction was completed by adding 0.5 µl of SuperScript II as described below, the concentration of each reagent that is present in the master mix was: 50 mM Tris-HCl (pH 8.0), 75 mM KCl, 0.5 mM dNTP Solution Mix, 10 mM DTT, 2% (v/v) deionized formamide (Millipore), 10 ng/µl ET SSB (New England Biolabs), 3 mM MnCl$_2$ (Fisher Scientific), 0.1% (v/v) Triton X-100, and 5 U/µl SuperScript II (Invitrogen). As described in the original SHAPE-MaP procedure, it is crucial to add MnCl$_2$ stock to the master mix immediately before performing reverse transcription because the manganese will begin to oxidize and precipitate in this solution[76]. Samples were placed on a pre-heated 42 °C thermal cycler for 2 min before 0.5 µl of SuperScript II was added to complete the master mix. The reverse transcription reaction was incubated at 42 °C for 50 min, and then at 70 °C for 15 min to heat inactivate SuperScript II. Samples were cooled to 12 °C, 1.25 U of RNase H (New England Biolabs) and 12.5 U of RNase I$_f$ (New England Biolabs) were added to each sample, and the samples were incubated at 37 °C for 20 min and then at 70 °C for 20 min to heat inactivate the RNases. The samples were briefly spun down in a mini-centrifuge and placed on a magnetic stand. The supernatant was aspirated and discarded, and the bead pellet was washed with 75 µl Storage Buffer (10 mM Tris-HCl (pH 8.0) and 0.05% (v/v) Triton X-100). The beads were resuspended in 25 µl of Storage Buffer, transferred to screw-cap tubes with an O-ring, and stored at −20 °C.

### Chemical probing of equilibrium-refolded intermediate transcripts

Single-round in vitro transcription using randomly biotinylated ppGpp template DNA was performed as described for TECprobe-VL in the section *Single-round* in vitro *transcription for TECprobe-VL and TECprobe-SL* and transcription was stopped by adding 150 µl of TRIzol LS. Intermediate transcripts were then purified as described in the section *RNA purification* except that volumes of chloroform and isopropanol were doubled to account for the increased sample volume and pellets were resuspended in 30 µl of 10 mM Tris-HCl (pH 8.0). Samples were placed in a heat block set to 95 °C for 2 min and then snap-cooled on ice for 1 min. 30 µl of 2X Equilibration Buffer (2X Transcription Buffer, 20 mM MgCl$_2$) containing either no ligand or 500 µM ppGpp was mixed with each sample and incubated at 37 °C for 20 min. Samples were then probed with BzCN as described in the section *RNA chemical probing*, purified by TRIzol LS extraction as described in the section *RNA purification*, and resuspended in 5 µl 10% (v/v) DMSO. All downstream sample processing steps were then followed exactly as described for TECprobe-VL experiments.

### Validation of dsDNA purification using homemade SPRI beads

To assess the effect of SPRI bead ratio on dsDNA recovery, increasing volumes of SPRI beads were mixed with 1 µl of a 4-fold dilution of 100 bp DNA ladder (New England Biolabs) and diluted to a final volume of 100 µl with 10 mM Tris-HCl (pH 8.0). Samples were incubated at room temperature for 5 min and purified as described in the section *SPRI bead purification of DNA*. DNA was eluted by resuspending beads in 10 µl of 10 mM Tris-HCl (pH 8.0), allowing the sample to sit undisturbed for 3 min, placing the tube on a magnetic stand for 1 min so that the beads collected on the tube wall, and transferring the supernatant into a microcentrifuge tube that contained 2 µl 6X DNA Loading Dye [10 mM Tris-HCl (pH 8.0), 30% (v/v) glycerol, 0.48% (w/v) SDS, 0.05% (w/v) Bromophenol Blue]. Samples were run on native TBE-polyacrylamide gels, stained with 1X SYBR Gold Nucleic Acid Stain in 1X TBE, and scanned on an Azure Biosystems Sapphire Biomolecular Imager using the 488 nM/518BP22 setting.

### Optimization of single-stranded nucleic acid purification using homemade SPRI beads

Single-stranded nucleic acid purification using SPRI beads was optimized by assessing the effect of isopropanol and PEG 8000

concentration on fragment size recovery. A mixture containing 1 µl of a 4-fold dilution of Low Range ssRNA Ladder (New England Biolabs) and 0.75 pmol each of RPIX_SC1_Bridge, dRP1_NoMod.R, and PRA1_shrt.F oligos (Supplementary Table 1) was combined with 22.5 µl SPRI beads, variable volumes of anhydrous isopropanol and 50% (w/v) PEG 8000, and RNase-free water to 100 µl. Samples were incubated at room temperature for 5 min and purified as described in the section *SPRI bead purification of DNA* above, except the beads were washed with 80% (v/v) ethanol instead of 70% (v/v) ethanol. Nucleic acids were eluted by resuspending the beads in 5 µl of 10 mM Tris-HCl (pH 8.0), allowing the tube to sit undisturbed for 3 min, placing the tube on a magnetic stand for 1 min so that the beads collected on the tube wall, and transferring the supernatant into a microcentrifuge tube that contained 15 µl of formamide loading dye. Samples were then analyzed as described in the section *Denaturing Urea-PAGE*.

### Reverse transcription efficiency assay

Reverse transcription (RT) efficiency assays that tested the ability of betaine, formamide, and ET SSB to eliminate primer dimer and promote full length cDNA synthesis were performed as follows: a *Cbe pfl* ZTP riboswitch DNA template lacking the terminator poly-U tract and containing an internal biotin-TEG modification was transcribed and RNA was purified as described in the sections *Single-round in vitro transcription for TECprobe-VL and TECprobe-SL* and *RNA purification* using the TECprobe-SL procedure, except that transcription volumes were 25 µl and chemical probing was not performed. The dRP1_5Bio.R oligo was annealed to the 3′ end of the RNA and immobilized on equilibrated Streptavidin C1 beads as described in the section *cDNA synthesis and cleanup*. Reverse transcription was then performed using either SuperScript II or SuperScript III as described below.

All SuperScript II reactions contained 50 mM Tris-HCl (pH 8.0), 75 mM KCl, 0.5 mM dNTP Solution Mix, 10 mM DTT, 3 mM MnCl$_2$, 0.1% (v/v) Triton X-100, and 5 U/µl SuperScript II. The beads were resuspended in 14.5 µl of SuperScript II master mix, which omitted RT additives and SuperScript II, mixed with 5 µl of an additive solution (described below), and pre-heated at 42 °C in a thermal cycler for 2 min before 0.5 µl of SuperScript II was added to the reaction.

All SuperScript III reactions contained 50 mM Tris-HCl (pH 8.0), 75 mM KCl, 0.5 mM dNTP Solution Mix, 5 mM DTT, 3 mM MnCl$_2$, 0.1% (v/v) Triton X-100, and 5 U/µl SuperScript III. The beads were resuspended in 15 µl of SuperScript III master mix and then mixed with 5 µl of an RT additive solution.

When present, betaine was included at 1.25 M, formamide was included at 2% or 5% (v/v), and ET SSB was included at 10 ng/µl. All reverse transcription reactions were incubated at 42 °C for 50 min, then at 70 °C for 15 min to heat inactivate reverse transcriptase, and cooled to 12 °C. Samples were briefly spun down in a mini centrifuge, placed on a magnetic stand, and the supernatant was aspirated and discarded. The beads were resuspended in 25 µl of Bead Elution Buffer [95% (v/v) formamide and 10 mM EDTA (pH 8.0)], heated at 100 °C for 5 min, placed on a magnetic stand, and the supernatant was collected and added to 125 µl of Stop Solution [0.6 M Tris-HCl (pH 8.0) and 12 mM EDTA (pH 8.0)]. Reactions were processed for denaturing urea-PAGE by phenol-chloroform extraction and ethanol precipitation. Briefly, 150 µl of Phenol-Chloroform-Isoamyl Alcohol (25:24:1) (ThermoFisher) was added to the reaction and thoroughly vortexed. The sample was centrifuged at 18,500 x g and 4 °C for 5 min and the aqueous phase was transferred to a new tube and precipitated by adding 15 µl of 3 M Sodium Acetate (pH 5.5), 450 µl of 100% Ethanol, and 1.5 µl of Glycoblue Coprecipitant and stored at −20 °C overnight. The sample was centrifuged at 18,500 x g and 4 °C for 30 min and the supernatant was aspirated and discarded. Pellets were resuspended with 15 µl of formamide loading dye and analyzed as described in the section *Denaturing Urea-PAGE*.

## Intermediate fraction analysis

Visualization of RNA-to-cDNA processing was performed by collecting intermediate sample fractions for analysis by denaturing PAGE. For this analysis, individual samples were processed in parallel to maintain the sample volumes used in the final procedure. Briefly, terminally biotinylated *Cba* ppGpp DNA template was transcribed in the absence of ligand and purified as described in the section *Transcription antitermination assays*. One sample was resuspended with 15 µl of formamide loading dye and the remaining samples were ligated to the 9N_VRA3 3′ adapter as described in the *RNA 3′ adapter ligation* section. Following ligation, one sample was TRIzol extracted and the pellet was resuspended with 15 of µl formamide loading dye. The remaining samples were purified as described in the section *SPRI bead purification of RNA*. One sample was TRIzol extracted after each SPRI bead purification (for a total of two samples extracted), and the pellets were resuspended in 15 µl of formamide loading dye. The remaining samples were annealed to dRP1_5Bio.R oligo and immobilized on equilibrated Streptavidin C1 beads as described in the section *cDNA synthesis and cleanup*. Samples were briefly spun down in a mini centrifuge and placed on a magnetic stand to dispose of the supernatant. The beads from one sample were resuspended in 25 µl of Bead Elution Buffer, heated at 100 °C for 5 min, placed on a magnetic stand and the supernatant was collected and added to 125 µl of Stop Solution. The remaining samples were reverse transcribed and purified as described in the *cDNA synthesis and cleanup*, except that the beads from one sample were resuspended in 25 µl of Bead Elution Buffer, instead of Storage Buffer, heated at 100 °C for 5 min, placed on a magnetic stand and the supernatant was collected and added to 125 µl of Stop Solution. The remaining reaction was washed with 75 µl of Storage Buffer, resuspended in 25 µl of Storage Buffer, and stored at −20 °C. Both samples in Stop Solution were phenol-chloroform extracted and ethanol precipitated, and the resulting pellets were each resuspended with 15 µl of formamide loading dye. The samples resuspended in formamide loading dye were analyzed as described in the section *Denaturing Urea-PAGE*. The sample resuspended in Storage Buffer was processed and analyzed as described in the *Test amplification of TECprobe libraries* section below except that serial 4-fold dilutions of bead-bound cDNA libraries were amplified for only 21 cycles.

## Test amplification of TECprobe libraries

The number of PCR amplification cycles needed for TECprobe libraries was determined by performing a test amplification adapted from Mahat et al.[77]. Briefly, 14 µl of PCR master mix was added to 6 µl of a 16-fold dilution of the bead-bound cDNA libraries, such that the final concentration of components in the 20 µl PCR were: 1X Q5 Reaction Buffer (New England Biolabs), 1X Q5 High GC Enhancer (New England Biolabs), 200 µM dNTP Solution Mix, 250 nM RPIX Forward Primer (Supplementary Table 1) 250 nM dRP1_NoMod.R Reverse Primer (Supplementary Table 1), 10 nM RPIX_SC1_Bridge (Supplementary Table 1), and 0.02 U/µl Q5 High-Fidelity DNA Polymerase. Amplification was performed for 21 or 25 cycles, at an annealing temperature of 62 °C and an extension time of 20 s. 20 µl of each supernatant was run on native TBE-polyacrylamide gels to assess both the fragment size distribution of the libraries and to determine the appropriate number of cycles for amplification of libraries for high-throughput sequencing. Representative gel images are shown in Supplementary Fig. 4c, d. Uncropped source gels are available in Supplementary Fig. 24.

## Preparation of dsDNA libraries for sequencing

Amplification of cDNA libraries for high throughput sequencing was performed by preparing separate 50 µl PCRs for each (+) and (-) sample that contained 1X Q5 Reaction Buffer, 1X Q5 High GC Enhancer, 200 µM dNTP Solution Mix, 250 nM RPI Indexing Primer (Supplementary Table 1), 250 nM dRP1_NoMod.R Reverse Primer (Supplementary Table 1), 10 nM SC1Brdg_MINUS or SC1Brdg_PLUS channel barcode

oligo (Supplementary Table 1), 12 µl of bead-bound cDNA library, and 0.02 U/µl Q5 High-Fidelity DNA Polymerase. Amplification was performed as indicated above, using the number of cycles determined by the test amplification. Supernatants from completed PCRs were each mixed with 100 µl of SPRI beads and purified as described in *SPRI bead purification of DNA*. DNA was eluted into 20 µl of 10 mM Tris-HCl (pH 8.0), mixed with 40 µl of SPRI beads, and purified as described in *SPRI bead purification of DNA* a second time. Twice-purified DNA was eluted into 10 µl of 10 mM Tris-HCl (pH 8.0) and quantified using the Qubit dsDNA HS Assay Kit (Invitrogen) with a Qubit 4 Fluorometer. Molarity was estimated using the length distribution observed during test amplification.

## High-throughput DNA sequencing

Sequencing of chemically probed libraries was performed by Novogene Co. on an Illumina HiSeq 4000 System using 2 × 150 PE reads with 10% PhiX spike in. TECprobe-VL libraries were sequenced at a depth of ~30 to ~60 million PE reads. TECprobe-SL libraries were sequenced at a depth of ~1 to ~2 million PE reads.

## Sequencing read pre-processing using cotrans_preprocessor

After comparing replicate data sets that were analyzed individually (Fig. 2e-g and Supplementary Fig. 1d-f), TECprobe-VL replicate data were concatenated and analyzed together. cotrans_preprocessor handles target generation, sequencing read pre-processing, and ShapeMapper2 run script generation for TECprobe-VL and TECprobe-SL data analysis. Source code and documentation for cotrans_preprocessor are available at https://github.com/e-strobel-lab/TECtools/releases/tag/v1.0.0.

## Target generation for TECprobe-VL and TECprobe-SL experiments using cotrans_preprocessor

For TECprobe-VL experiments, two types of sequence targets (3′ end targets and intermediate transcript targets) were generated by running cotrans_preprocessor in MAKE_3pEND_TARGETS mode. 3′ end targets files contained the 3′-most 14 nt of every intermediate transcript and all 1 nt substitution, insertion, and deletion variants of these sequences. The 3′ end targets file is used to demultiplex fastq files based on intermediate transcript identity, which is inferred from the RNA 3′ end, in the section *Sequencing read preprocessing for TECprobe-VL experiments*, below. The accuracy of 3′ end mapping was assessed by generating native and randomized variant test data using the options –T or –T –R –U 30, respectively, and processing the test data as described below in the section *Sequencing read preprocessing for TECprobe-VL experiments*. Based on these analyses, the 14 nt default length of the 3′-end target sequences was sufficient for accurate intermediate transcript identity determination. Intermediate transcript targets comprise individual fasta files for every intermediate transcript sequence with an $A_{+1}T_{+2}$ dinucleotide and the SC1 adapter appended to the 5′ end in lower case, so that these sequences are excluded from ShapeMapper2 analysis. For TECprobe-SL experiments, a single target was generated by running cotrans_preprocessor in MAKE_SINGLE_TARGET mode.

## Sequencing read pre-processing for TECprobe-VL experiments

For TECprobe-VL experiments, cotrans_preprocessor manages adapter trimming using fastp[78] and demultiplexes aggregate sequencing reads by modified and untreated channel and by intermediate transcript identity. It was necessary to demultiplex aggregate sequencing reads by intermediate transcript identity, which is inferred from the RNA 3′ end, so that each intermediate transcript could be analyzed separately by ShapeMapper2[79]; this avoids sequencing read multimapping during bowtie2 alignment. TECprobe-VL sequencing reads were processed by running cotrans_preprocessor in PROCESS_MULTI mode, which performs the following operations: First, fastp is called to

perform adapter trimming and to extract unique molecular index and channel barcode sequences from the head of reads 1 and 2, respectively. After fastp processing is complete, cotrans_preprocessor splits the fastp output files by channel (modified barcode [RRRYY] or untreated barcode [YYYRR]) and intermediate transcript identity, and generates the files `smooth_transition.sh` and `config.txt`. `smooth_transition.sh` is a shell script that can be used to apply neighboring transcript smoothing by generating fastq files in which, for each intermediate transcript *n*, sequencing reads for intermediate transcripts *n-1*, *n*, and *n+1* are concatenated into a single file (excluding the minimum transcript *min*, which is concatenated as *min*, *min+1*, and the maximum transcript *max*, concatenated as *max-1*, *max*). `config.txt` is a configuration file that is used to specify information about a data set when generating a ShapeMapper2[79] run script as described below in the section *ShapeMapper2 run script generation using cotrans_preprocessor*.

### Sequencing read pre-processing for TECprobe-SL experiments

For TECprobe-SL experiments, cotrans_preprocessor manages adapter trimming using fastp[78] and demultiplexes aggregate sequencing reads by modified and untreated channels. TECprobe-SL sequencing reads were processed by running cotrans_preprocessor in `PROCESS_SINGLE` mode, which performs the following operations: First, fastp is called to perform adapter trimming and to extract the channel barcode sequence from the head of read 2. After fastp processing is complete, cotrans_preprocessor splits the fastp output files by channel (modified barcode [RRRYY] or untreated barcode [YYYRR]) and generates the file `config.txt`. As described above in the section *Sequencing read pre-processing for TECprobe-VL experiments*, `config.txt` is a configuration file that is used to specify information about a data set when generating a ShapeMapper2[79] run script.

### ShapeMapper2 run script generation using cotrans_preprocessor

Shell scripts to run ShapeMapper2[79] analysis were generated by running cotrans_preprocessor in `MAKE_RUN_SCRIPT` mode. For TECprobe-SL experiments, the run script contained a single command. For TECprobe-VL experiments, the run script contained a command for every intermediate transcript. The following Shape-Mapper2 options were used: `-min-depth 500` was used to ensure that all data exceeded the minimum cutoff so that sequencing depth filtering could be performed separately during heatmap generation. `-min-qual-to-trim 10` was used to keep read pairs in which read 2 contained low-quality base calls near the read head. `-min-qual-to-count 25` was used to filter low-quality base calls during reactivity calculation.

### TECprobe-VL reactivity heatmap generation

ShapeMapper2 output data for each individual transcript in a TECprobe-VL experiment were assembled into a single matrix in csv format by the `compile_SM2_output` script. Reactivity and read depth data matrices and heatmaps were then generated by the `generate_cotrans_heatmap` script. Some transcript lengths were not enriched because the template DNA strand segment composed of the HP4_5bio.R primer does not contain randomly positioned internal biotin modifications. These transcripts were excluded from the heatmaps and all analyses. Source code and documentation for `compile_SM2_output` and `generate_cotrans_heatmap` are available at https://github.com/e-strobel-lab/TECprobe_visualization.

### Generation of correlation plots for TECprobe-VL data sets

Reactivity matrices generated by the `generate_cotrans_heatmap` script were used to generate replicate correlation plots using the `plot_cotrans_correlation` script. Transcript lengths that were not enriched by biotin-streptavidin roadblocking (as described above in *TECprobe-VL reactivity heatmap generation*) were excluded from the analysis. Nucleotides at RNA 3' ends with a reactivity value of zero were excluded from the analysis. If the reactivity of a nucleotide in one replicate was NaN, the corresponding reactivity in the other replicate was masked as NaN. Source code and documentation for `plot_cotrans_correlation` are available at https://github.com/e-strobel-lab/TECprobe_visualization.

### Comparison of reactivity and background mutation rates for neighboring transcript lengths

The correlation of reactivity values and background mutation rates for neighboring transcripts was determined using the `compare_cotrans_neighbors` script. Pearson's correlation coefficients for reactivity and background mutation rate were calculated for each pair of neighboring transcripts; the 3'-most nucleotide of the longer transcript, which is not present in the shorter transcript, was omitted from the analysis. Transcript lengths that were not enriched by biotin-streptavidin roadblocking (as described above in *TECprobe-VL reactivity heatmap generation*) were excluded from the analysis. The set of correlation coefficients for each individual TECprobe-VL experiment were visualized using violin plots. Source code and documentation for `compare_cotrans_neighbors` are available at https://github.com/e-strobel-lab/TECprobe_visualization.

**Modeling and visualization of RNA secondary and tertiary structures.** Non-native intermediate ppGpp riboswitch structures were modeled as follows: For each transcript of interest, the 3'-most 12 to 14 nt were forced to be single-stranded to account for the *E. coli* RNAP footprint and possible base pairing configurations for the remaining transcript were generated using the RNAstructure v6.4[49] stochastic command with default settings. From this set of structures, putative folding intermediates were manually selected based on the agreement of the predicted structure with reactivity data and by assessing which nucleotides had recently emerged from RNAP at the transcript length when the intermediate is first observed and when the intermediate rearranges into a subsequent structure. The holo ppGpp aptamer structure was modeled by manually assessing the agreement of holo ppGpp aptamer reactivity profiles with conserved secondary and tertiary structure elements that were identified crystal structures[47,48]. The apo ppGpp aptamer, for which a crystal structure does not exist, was modeled by identifying ppGpp-dependent differences between the apo and holo ppGpp aptamer reactivity profiles and determining the interactions that these nucleotides make within the crystal structures; in all cases, DMS reactivity increased in the absence of ppGpp which indicates exposure of the Watson-Crick face and implies that a pairing interaction observed in the holo aptamer is either transient or non-existent in the apo aptamer. Crystal structures were visualized using UCSF Chimera[80].

**RNA structure prediction.** RNA structure prediction was performed using the RNAstructure v6.4[49] Fold command with default settings. Putative ppGpp riboswitches were selected by filtering ykkC-variant riboswitches identified by Nelson et al.[44] for sequences that match the ppGpp riboswitch consensus sequence and which do not contain a guanosine that is highly conserved in phosphoribosyl pyrophosphate (PRPP) riboswitches (position 90 in the *Cba* ppGpp riboswitch). This may exclude some ppGpp riboswitches, but ensures that PRPP riboswitches are excluded. The aptamer segments used for structure prediction started 1 nucleotide upstream of the conserved aptamer sequence and extended either 19 or 38 nucleotides downstream. Sequence randomization was performed as follows[33]: unbiased randomization allowed an equal probability for all nucleotides at each position, wild-type nucleotide-biased randomization was performed using the observed nucleotide frequency for each position in the ppGpp riboswitch multiple sequence alignment, and shuffled

randomization was performed by randomly reordering the nucleotides of natural sequences.

### Reporting summary

Further information on research design is available in the Nature Portfolio Reporting Summary linked to this article.

## Data availability

The raw sequencing data generated in this study have been deposited in the Sequencing Read Archive (https://www.ncbi.nlm.nih.gov/sra) with the BioProject accession code PRJNA929456. Individual BioSample accession codes are available in Supplementary Table 4. The processed reactivity data have been deposited in the RNA Mapping Database[81] (https://rmdb.stanford.edu/). Individual accession codes for each data set are available in Supplementary Table 5. The ShapeMapper2 output files for these data have been deposited in Zenodo (DOI: 10.5281/zenodo.7640593). The TIFF images of all gels generated in this study have been deposited in Zenodo (DOI: 10.5281/zenodo.10041572). Source data are provided with this paper. The crystal structures of the *T. carboxydivorans* ZTP, *T. petrophila* fluoride, *T. mathranii* PRPP G96A (ppGpp-binding), and *Sulfobacillus acidophilus* DSM 10332 riboswitches used in this study are available in the RSCB Protein Data Bank under the accession codes 4ZNP, 4ENC, 6CK4, and 6DMC. Cotranscriptional SHAPE-Seq Data for the ZTP and fluoride riboswitches used in this study was downloaded from the source data associated with the original publications. Source data are provided with this paper.

## Code availability

TECtools can be accessed at https://github.com/e-strobel-lab/TECtools/releases/tag/v1.0.0[82]. Scripts used for data visualization can be accessed at https://github.com/e-strobel-lab/TECprobe_visualization[83]. The script used for RNA structure prediction can be accessed at https://github.com/e-strobel-lab/Publications.

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

## Acknowledgements

This work was supported by the National Institute of General Medical Sciences of the National Institutes of Health under Award Number R35GM147137 (to E.J.S) and by start-up funding from the University at Buffalo (to E.J.S). The content is solely the responsibility of the authors and does not necessarily represent the official views of the National Institutes of Health.

## Author contributions

E.J.S., conceptualization; C.E.S. and E.J.S., methodology; C.E.S., investigation; C.E.S. and E.J.S, Validation; C.E.S. and E.J.S, Formal Analysis; C.E.S. and E.J.S, Software; C.E.S. and E.J.S, writing – original draft; C.E.S. and E.J.S, writing – review & editing; E.J.S., supervision; E.J.S., funding acquisition.

## Competing interests

The authors declare no competing interests.
