## [Peer Review File · Nature Communications]

Observation of coordinated RNA folding events by systematic cotranscriptional RNA structure probingREVIEWER COMMENTS

Reviewer #1 (Remarks to the Author):

Szyjka and Strobel describe a new chemical probing strategy named TECprobe-ML for measuring RNA cotranscriptional folding pathways. The primary innovation of TECprobe-ML is to modernize cotranscriptional methods to use MaP reverse transcription readout rather than termination. They also describe several modest improvements to MaP RT, streamlining of sample preparation, and data analysis. They then validate their method by repeating characterization of the previously studied ZTP and fluoride riboswitches, and then provide new data for the ppGpp system. Overall, the data collected appears to be of very high quality and the study is rigorous. The collective methodological advances, while somewhat obvious, are important and will drive the cotranscriptional folding field forward. Nonetheless, given that cotranscriptional folding remains a niche field, the broader impact of these methodological advances is unclear. The presentation of the riboswitch data is very dense, reducing the impact of these studies for a broader audience.

Major critiques

-The authors argue with ZTP and fluoride riboswitch experiments that their method both recapitulates prior observations while also allowing visualization of previously unobserved events. However, this is told in words, never clearly shown in figures. Head-to-head comparisons with prior results (e.g. they get very similar looking matrices) would help prove their argument that their (easier) method supplies the same measurements, and would also help highlight the gaps their new measurements are able to fill.

-The data presentation in the main text and in Figures 3-7 is very dense and shows numerous, very similar looking plots. This makes it hard to find and appreciate the important / novel results. Key arguments about coordinated folding events, such as the pseudoknot in the ZTP system, are hard to evaluate when the data are not directly juxtaposed but rather split across multiple panels that must be cross-referenced and compared. The manuscript would benefit from making the figures more concise, emphasizing the novel claims while data confirming prior observations is placed in the SI.

-It is hard to discern the folding pathways from the structural diagrams shown in the figures. For example, what are the structures of the antiterminators for the three riboswitches? For the ppGpp system, which the authors are first to characterize, where is a clear summary of the folding trajectory, notating the key folding steps/intermediates from beginning to end? Figure 6 shows early folding, but not late folding events.

-The discussion fails to provide any broader context for “what was learned” from the ppGpp system and whether these lessons can more broadly inform understanding of cotranscriptional folding.

-The authors collect unique parallel BzCN and DMS datasets on the same systems. However, with the exception of 1-2 short statements, they provide minimal discussion / comparison of the data obtained from these experiments. For example, one notable difference that goes unmentioned is that DMS

experiments are done on a much longer time-scale and under multiple hit conditions compared to BzCN. Does this have any impact on the observed folding pathways? Figure 6 ostensibly compares BzCN and DMS, but then focuses on different subsets of nucleotides. Overall, providing more insight into the strengths and weaknesses of these reagents would help elevate the impact of the work and help the field know which reagents to select for future studies.

Minor comments:

The title is very generic – a more specific title making clear that this paper is presenting a new method would be more informative.

TECprobe-ML is somewhat of an awkward name. I recommend not using “ML”, since this will make many readers think of machine learning.

Page 3: The authors state “This ligation is prone to sequence bias that distorts the reactivity measurement.” Can the authors provide a reference here, as the relevance of bias is controversial.

Can the authors provide more detail on the rationale behind the formamide and SSB additions to RT? For example, relevant citations? Are these additions likely to more broadly improve MaP? Also, looking at figure S3b, it is not clear that formamide has an impact relative to SSB.

Reviewer #2 (Remarks to the Author):

Szyjka and Strobel present an adapted RNA structure probing method for examining the folding landscape of RNAs during transcription called TECprobe-ML. This approach has several advantages over the original co-transcriptional SHAPE-seq: 1) Users can directly amplify the cDNA by priming on a structured 5' cassette instead of ligating an adapter during library preparation. 2) The method now uses mutational profiling which has already been established in other sequencing-based RNA structure probing methods. 3) The authors develop a method to smooth the probing data. The authors show that this approach resolves structures close to the 5' and 3' end of the transcript, which were previously difficult or impossible to detect. The authors use TECprobe-ML to examine the folding pathway of two riboswitches that have been previously characterized by similar methods, the ZTP and fluoride riboswitches, and investigate the folding pathway of the ppGpp riboswitch, which has not been previously characterized. I think the work is thorough and highlights that the method has been optimized from the previous version. I have a few comments:

Major comments:

1. It is difficult to evaluate the comparison to the previous iteration of co-transcriptional SHAPE-seq. The authors make a series of claims in the validation sections to show that TECprobe-ML performs better than the original co-transcriptional SHAPE-seq, but it is difficult to determine whether these are supported by their data because there are no direct figure comparisons with the previous work. My concern stems from the fact that the data is different between the two studies and this makes it very difficult for a reader to compare between the two papers. For example, the reactivity (measured by RT stops) values range from 0 - 2 in the Strobel et al. 2019 paper, whereas the reactivities range from 0 - 0.01 in this manuscript (measured by MaP). Could the authors include a side-by-side comparison of particular nucleotides or transitions with relative reactivities to highlight the improvement over the first method? Are there quantitative metrics that the authors can use to describe how much better TECprobe-ML performs?

2. In the validation sections, it is sometimes not clear in the text what is new for the TECprobeML and what is already known from previous results. For example, in the ZTP riboswitch section, are the changes seen in the linker between P1/P2 and P3 new results for this method? Similarly, are the coordinated changes in the PK region completely new? How was this detected previously? Please clarify in the text.

3. The authors performed full-length structure analysis (TECprobeSL) for the ZTP riboswitch in Supplemental Figure S8. Have the authors done this analysis with different 5' cassette sequences to confirm that it does not interfere with overall folding? I think the functional assays are useful and convincing in this case, but I think it would be useful to show that the full-length structure is also not perturbed by the incorporation of SC1 or other hairpins. This would be important for application to studying the folding of other RNAs which do not have a simple functional assay.

4. In the discussion, would the authors expand on why improving resolution near the 3' end is important for studying folding of other RNAs in order to highlight the significance and applications of the improved method.

5. Would the authors describe how this method compares to other RNA-seq based structure probing methods such as SPET-seq?

6. What is the outlook for expanding upon this method? Can this approach be combined with other MaP-based structure probing approaches that examine RNA structural changes upon protein binding or examine the structural heterogeneity?

Minor comments:

1. With regards to performance, in Figure 2a – what are the alignment rates for the original co-transcriptional SHAPE-seq?

2. Figure 2b-d – Why are there large regions in all three cases that are not enriched with the biotin-streptavidin roadblocks? How is this determined?

3. Can the authors comment on whether the SC1 cassette is needed if the addition of ET SSB solves the primer dimer issue as in Figure S3b? Can you add SSB to the cDNA amplification step?
4. The authors should add the descriptions of the color coding (i.e. in the ZTP riboswitch section, green/purple for 0 and 1 mM ZMP) to all the figure legends for clarity.
5. For the ppGpp riboswitch, the authors should include a secondary structure diagram of the complete ppGpp riboswitch they tested (transcript 134 I believe). They discuss late folding events that occur with transcripts at 108 and 115 nts but there is no complete structure shown in Figure 7 or Figure S13.
6. In Figure 6k and l, I think the labels for P2a and P2b are shifted. P2a should be near the base of the coaxial stack.

Reviewer #3 (Remarks to the Author):

RNAs fold during transcription and this is important for their regulation because the timing of hairpin formation can regulate the function of genes. In this manuscript, Szyjka and Strobel develop and optimize a method for probing co-transcriptional folding that they call “TECprobe-ML”. This study represents both method development and key biological insights into principles of RNA folding and riboswitch gene regulation. TECprobe-ML is an improvement over their previous method of “Co-transcriptional SHAPE-seq” and they go on to clearly list all the improvements. The authors probe the folding of a set of riboswitches using this method, first validating it on the ZTP and fluoride riboswitches and then extending it to the ppGpp riboswitch. Validation sections, which are critical parts of any methods development section of a manuscript, read very well and provide new insights, including information on 3'-proximal residues and read-through of very stable RNA structures such as terminators. There are nice controls that addition of a 5'-element does not affect response of antitermination to ligand. Results on the three riboswitches support formation of key helices in antitermination. This is an important paper because it can aid modeling and prediction of co-transcriptional folding of RNA, which is understudied and not well-understood but is prevalent and dominates the folding of many RNAs. I ask the authors to address the following points.

1. Having “ML” in a method title immediately implies machine learning. The authors intend for it to stand for “multilength”. I strongly encourage them to choose a different abbreviation to avoid confusion.
2. The first section of the Results, which explains the TECprobe-ML, is missing any description of the “biotin-streptavidin” roadblock depicted in Figure 1. Presumably this roadblock got introduced (randomly?) in solid-phase synthesis of the BS DNA or during a PCR of the dsDNA template. Ref 15 suggests that the PI developed this in the Lucks lab but at least some description is needed here.
3. Data smoothing across three lengths plays an important part in enhancing replication of the data. This is a purely computational change, however, and so one wonders if smoothing might have helped the original non-MaP approach.
4. Weeks and Mustoe have turned to DMS mapping over SHAPE, at least for PAIR-MaP that uses single-molecule correlated chemical probing. This is because DMS is much more sensitive than SHAPE for

mutational profiling, as established by Rouskin. This requires pH 8 to hit all four bases, but transcription is often fine at this pH. Have the authors considered switching to DMS-based method over SHAPE? Clearly, they can use DMS, as evidenced by SI figures with DMS probing for A/C on the ZTP and fluoride riboswitches.

5. p6. J1/2. It would be helpful to annotate Fig 3e with J1/2 on the appropriate nucleotides on the x-axis as was done for other elements. The authors should check all their writing and figures for such omissions and put in annotations to help the reader.

6. p9. It is interesting and at times convincing to read about the intermediate hairpins (IH) that precede formation of P2 in the ppGpp riboswitch. Given that P2 is a long element, this is not surprising as it is hard to imagine it otherwise given RNA's proclivity to fold; nonetheless, it is impressive to see. This raises a deeper question: Do IH hairpins serve any purpose in guiding folding? For example, is there evidence for their conservation when doing sequence comparison across ppGpp riboswitches? See papers from Isambert and Siggia where such "folding guides" have been proposed for such His: Isambert, H. & Siggia, E. D. (2000). Modeling RNA folding paths with pseudoknots: Application to hepatitis delta virus ribozyme. *Proc. Natl. Acad. Sci. U S A* 97, 6515-6520. 18642

7. p9, para 2. There is a difference between equilibrated and co-transcriptional IH data with BzCN. How much would this change with pausing and native polymerases. (The authors use *E. coli* polymerase for a *Clostridiales* bacterium riboswitch.)

8. Figure 7a could be clearer. What are the dashed boxes for in 7a? Is this the "long-range interactions with the minor group of P2b"? Annotate in both 7a and 7b. Specifically, in 7b, annotate the various pairings. Other figures should be cross-checked for similar changes.

9. p11. The authors write, "In contrast, ppGpp-dependent reactivity signatures were not detectable in transcripts that accumulated at the terminal transcription roadblock downstream of the termination site (Figure 7 and Supplementary Figure 13h-l)." In Supp Fig 13h-l, do they want us to look at the little strip at 170? If so, it should be stated. I can see ppGpp-dependent reactivity signatures in transcripts downstream of termination for lengths 126-135 in this figure.

REVIEWER COMMENTS

Reviewer #1 (Remarks to the Author):

Szyjka and Strobel describe a new chemical probing strategy named TECprobe-ML for measuring RNA cotranscriptional folding pathways. The primary innovation of TECprobe-ML is to modernize cotranscriptional methods to use MaP reverse transcription readout rather than termination. They also describe several modest improvements to MaP RT, streamlining of sample preparation, and data analysis. They then validate their method by repeating characterization of the previously studied ZTP and fluoride riboswitches, and then provide new data for the ppGpp system. Overall, the data collected appears to be of very high quality and the study is rigorous. The collective methodological advances, while somewhat obvious, are important and will drive the cotranscriptional folding field forward. Nonetheless, given that cotranscriptional folding remains a niche field, the broader impact of these methodological advances is unclear. The presentation of the riboswitch data is very dense, reducing the impact of these studies for a broader audience.

Thank you for appreciating the rigor of the study and quality of the data, and for suggesting ways to improve the impact of the work for a broad audience.

Major critiques

1. The authors argue with ZTP and fluoride riboswitch experiments that their method both recapitulates prior observations while also allowing visualization of previously unobserved events. However, this is told in words, never clearly shown in figures. Head-to-head comparisons with prior results (e.g. they get very similar looking matrices) would help prove their argument that their (easier) method supplies the same measurements, and would also help highlight the gaps their new measurements are able to fill.

We agree that the absence of direct comparisons to previous data was a critical omission in the original manuscript. To address this, the manuscript now includes seven new supplementary figures in which we directly compare new BzCN probing data with data collected by cotranscriptional SHAPE-Seq. In Supplementary Figures 6 and 10, TECprobe and cotranscriptional-SHAPE-seq reactivity matrices are juxtaposed to show that the overall reactivity trends and observed folding events are the same for both methods. In Supplementary Figures 7, 8, 9, 11, and 12, TECprobe and cotranscriptional-SHAPE-seq reactivity trajectories for specific nucleotides that are referenced in the text are juxtaposed. These figures show that although overall trends are the same for both methods, critical information is either missing or not interpretable in the cotranscriptional SHAPE-Seq data. For example, Supplementary Figure 8c shows that ZMP-dependent decreased reactivity at U94 and G95 of the *pfl* ZTP aptamer, which directly contact ZMP, is not detectable using cotranscriptional SHAPE-Seq. Similarly, Supplementary Figure 11e shows that fluoride-dependent formation of the linchpin base pair, which renders the fluoride aptamer resistant to terminator base pair propagation, was not detectable using cotranscriptional SHAPE-Seq. We believe that these new figures provide a thorough comparison of the methods that will enable readers to assess the performance improvements made by TECprobe.

2. The data presentation in the main text and in Figures 3-7 is very dense and shows numerous, very similar looking plots. This makes it hard to find and appreciate the important / novel results. Key arguments about coordinated folding events, such as the pseudoknot in the ZTP system, are hard to evaluate when the data are not directly juxtaposed but rather split across multiple panels that must be cross-referenced and compared. The manuscript would benefit from

making the figures more concise, emphasizing the novel claims while data confirming prior observations is placed in the SI.

Thank you for this suggestion. We have made extensive changes to Figures 3, 4, 7 and several Extended Data (formerly Supplementary) Figures to address this. In all figures, we have reorganized the panels to facilitate comparisons that are made in the text. For example, in Figure 3, panel e now compares reactivity trajectories that show the rearrangement of the intermediate hairpin IH1 when the P1 subdomain of the *pfl* ZTP aptamer folds. Similarly, panel f compares reactivity trajectories associated with pseudoknot folding and ligand binding. In both cases, three of the four plots that are shown support new findings. To enable this, we moved all plots that showed known ZMP-dependent reactivity changes to Extended Data Figure 3, which is arranged to facilitate comparisons between BzCN and DMS data. Although we did not remove plots from Figures 4 or 7, we simplified and reorganized the panels to facilitate evaluation of the coordinated folding events that are described in the text. Additionally, we revised all Extended Data Figure panels that could benefit from being reorganized in this way. Overall, these changes have made it significantly easier to visualize coordinated folding events.

3. It is hard to discern the folding pathways from the structural diagrams shown in the figures. For example, what are the structures of the antiterminators for the three riboswitches? For the ppGpp system, which the authors are first to characterize, where is a clear summary of the folding trajectory, notating the key folding steps/intermediates from beginning to end? Figure 6 shows early folding, but not late folding events.

We have added structures showing the full length ZTP (Extended Data Figure 2g), fluoride (Figure 3a), and ppGpp (Extended Data Figure 9c, g) riboswitches, and have also added text describing these structures in the relevant sections of the results. To summarize, the antiterminated structures of the ZTP and fluoride riboswitches contain hairpins that likely block terminator nucleation (in the case of the ZTP switch) or terminator base pair propagation (in the case of the fluoride switch) after antitermination occurs and the full terminator sequence has emerged from RNAP. In contrast, a predicted structure in the downstream terminator stem of the ppGpp riboswitch is unlikely to block terminator nucleation. This may explain why the ligand-bound ZTP and fluoride aptamers persist following antitermination but the ligand-bound ppGpp aptamer is eventually disrupted by the terminator hairpin.

We have also added a new main text figure that summarizes the folding pathway of the ppGpp riboswitch (Figure 8). This figure indicates when each folding intermediate is observed and when critical events, such as ppGpp binding and terminator nucleation, can occur. In addition, we have added a new results section (included below) that describes folding of the ppGpp aptamer P3 and P4 hairpins. When preparing this section, we realized that P3 base pairs become less reactive to DMS when the ppGpp aptamer four-way junction has folded, and that ppGpp binding further decreases the reactivity of these nucleotides. This suggests that ppGpp binding stabilizes the four-way junction structure, which may contribute to transcription antitermination by reducing the lability of P1.

“Folding of the P3 and P4 hairpins

In principle, P3 can fold once it has fully emerged from RNAP at transcript ~82. With the exception of A64, nucleotides within P3 are not reactive to BzCN and nucleotides U60 and A63 within the L3 are BzCN-reactive, suggesting that P3 does fold (Extended Data Figure 6a, b). In contrast, nucleotides A58 and C66 within P3 are consistently DMS reactive until after the full aptamer has emerged from RNAP at transcript ~118 (Extended Data Figure 6c, e). One possible explanation for this apparent contradiction is that P3 may not be stably folded until the

entire ppGpp aptamer has folded, and the longer reaction time used for DMS probing was able to detect the unfolded P3 stem while BzCN probing was not. Notably, ppGpp binding decreased the reactivity of P3 nucleotides, indicating that P3 is stabilized in the context of the full, ligand-bound aptamer (Extended Data Figure 6c, e). Regardless of when P3 is stably folded, P4 folding was detected as decreased BzCN reactivity at U72 and U73 from transcript 100 to 105, as the downstream P4 stem emerges from RNAP (Extended Data Figure 6c, e)."

4. The discussion fails to provide any broader context for "what was learned" from the ppGpp system and whether these lessons can more broadly inform understanding of cotranscriptional folding.

Thank you for this suggestion. We have added two new paragraphs in the discussion that provide broader context for what was learned from the ppGpp riboswitch system, which we have included below. The first paragraph focuses on what was learned regarding how ligand binding renders the ppGpp aptamer resistant to terminator hairpin folding and overlaps to some degree with our discussion of the usefulness of parallel BzCN and DMS probing in cotranscriptional RNA structure probing experiments. The second paragraph, which we believe is more broadly important, focuses on what was learned regarding transient non-native structure formation. As described in response to Reviewer 3's Comment 6, we assessed the capacity of 30 putative ppGpp riboswitches from bacterial genomes for intermediate hairpins using minimum free energy structure prediction. While IH1 may occur in many ppGpp riboswitch systems, IH2 and IH3 do not. Nonetheless (and perhaps unsurprisingly), all sequences that were tested have the capacity to form some non-native structure. Given that random RNA sequences tend to form structures (Schultes et al, 2005), we suggest that easily resolvable non-native structures may be a common occurrence in cotranscriptional RNA folding pathways even when no conserved non-native structure is present.

"The usefulness of parallel BzCN and DMS probing data sets was most evident in the detection of ppGpp binding by the *Cba* ppGpp riboswitch. In contrast to the *pfl* ZTP and *crcB* fluoride riboswitches, little evidence of ppGpp-dependent aptamer stabilization was detected by BzCN probing (Extended Data Figure 6). Upon complete aptamer folding, the BzCN reactivity of nucleotides within the ligand binding pocket decreased regardless of whether ppGpp was present. This indicates that the flexibility of nucleotides within the ligand binding pocket is approximately equivalent regardless of whether ppGpp is bound and that the ligand binding pocket is at least partially organized. However, DMS probing revealed that the G10-A68 pair and contacts between G48 and A70 do not stably form until C69 pairs with the G nucleobase of ppGpp. Together, these contacts directly stabilize P1 by extending its helical stack. DMS probing also revealed that ppGpp binding indirectly stabilizes distal contacts between P2b and P4 and several base pairs in P3. These ligand-dependent changes in structure presumably stabilize P1 indirectly by maintaining the four-way junction structure of the folded ppGpp aptamer. Taken together, these observations indicate that the apo ppGpp aptamer is folded into a four-way junction structure and suggest that ppGpp binding stabilizes P1 directly by extending the P1 stack and indirectly by stabilizing long-range contacts between P2 and P4, thereby conferring terminator resistance.

Analysis of ppGpp riboswitch folding using TECprobe-VL established a positional map of folding intermediates that can occur as the *Cba* ppGpp aptamer folds (Figure 8). Like the *pfl* ZTP aptamer, transient intermediate hairpins precede *Cba* ppGpp aptamer folding. The first transient structure, IH1, is potentiated by the high likelihood of forming contiguous base pairs between conserved poly-purine and poly-pyrimidine stretches within the ppGpp aptamer and therefore

may be present in other ppGpp riboswitch folding pathways. Minimum free energy structure predictions suggest that the subsequent transient structures that were observed for the *Cba* ppGpp riboswitch, IH2 and IH3, are not present in other ppGpp riboswitch folding pathways (Supplementary Figure 13). However, all putative ppGpp riboswitches that were assessed are predicted to form non-native structures that precede P2 hairpin folding. Determining whether the observed transient structures function as ‘folding guides’⁶⁶ that coordinate *Cba* ppGpp aptamer folding is not possible using the current data and will likely require approaches for evaluating how perturbing these structures affects riboswitch function. In addition to the *pfl* ZTP and *Cba* ppGpp riboswitches, transient non-native structures have also been detected within the folding pathways of the *E. coli* signal recognition particle, RNase P, tmRNA, and *thiB* riboswitch RNAs^{14,16,67,68}. In most of these cases, the non-native structure precedes folding of a native structure that requires base pairs between distal nucleotides. While some transient structures are conserved^{56,61}, others, such as the *pfl* ZTP riboswitch IH1 hairpin and *Cba* ppGpp riboswitch IH2 and IH3 hairpins, appear to be a consequence of the nascent transcript sampling energetically favorable structures until it is long enough that the native structure can fold. The occurrence of non-native structures in such contexts is not surprising given the capacity of random RNA sequences for the formation of secondary structures^{69,70}. In such cases, intermediate structures that can be easily resolved to permit the native structure to fold are likely advantageous. This is likely to be particularly important for riboswitches that regulate transcription because the resolution of non-native structures must occur cotranscriptionally.”

5. The authors collect unique parallel BzCN and DMS datasets on the same systems. However, with the exception of 1-2 short statements, they provide minimal discussion / comparison of the data obtained from these experiments. For example, one notable difference that goes unmentioned is that DMS experiments are done on a much longer time-scale and under multiple hit conditions compared to BzCN. Does this have any impact on the observed folding pathways? Figure 6 ostensibly compares BzCN and DMS, but then focuses on different subsets of nucleotides. Overall, providing more insight into the strengths and weaknesses of these reagents would help elevate the impact of the work and help the field know which reagents to select for future studies.

Thank you for this suggestion. We have revised the discussion to include a paragraph (included below) that compares BzCN and DMS probing data. To summarize, the same folding events were detectable using both probes, but in some cases the longer probing time used for DMS may allow RNAP to backtrack, thereby shifting the apparent length at which a folding transition occurs by several nucleotides. In several cases, divergent reactivity trends for the two probes corresponded to nucleotides in which backbone flexibility is reduced by stacking or by the formation of a Hoogsteen face base pair while the Watson-Crick face remains unpaired. Because the information obtained using the two probes can be complementary, we recommend collecting parallel BzCN and DMS data sets when possible. We also suggest that properties of DMS probing, such as high signal-to-noise and the ability to perform single-molecule correlated chemical probing, will be useful for cotranscriptional RNA structure probing experiments.

“The collection of BzCN and DMS data sets for three riboswitch systems enabled us to compare the performance of each probe in cotranscriptional RNA chemical probing experiments. Whereas the BzCN probing reaction was performed using single-hit conditions and self-quenches in ~1s, the DMS probing reaction was performed using multiple-hit conditions and was quenched after a 5 minute reaction. Despite the longer reaction time for DMS probing, the same folding intermediates were observed with both probes for all riboswitch targets. However, it is important to note that transcription was allowed to proceed for 2 min before BzCN or DMS

probing was started. As described further below, this could allow cotranscriptionally folded RNA that is displayed from RNAP equilibrate to some degree before the chemical probing reaction begins. This limitation cannot be circumvented because it is essential to let the single-round transcription reaction run to completion before chemical probing begins. One possible consequence of probing with DMS for several minutes is that static transcription complexes can potentially backtrack over time, which displaces the RNA 3' end from the RNAP active center. Because the location of RNAP is inferred from the RNA 3' end, backtracking can shift the apparent transcript length at which a folding transition occurs forward. While most folding transitions occurred at the same transcript length in both BzCN and DMS data sets, the decrease in reactivity of the IH1 loop and downstream P1 stem of the ZTP aptamer during the IH1-to-P1 folding transition reached its lowest point at transcript 80 with BzCN and transcript ~86 with DMS (Figure 3e and Extended Data Figure 2c). In several cases BzCN and DMS probing provided complementary information. For example, in the *pfl* ZTP riboswitch the stabilization of a Hoogsteen/Hoogsteen base pair between G19 and A47 was detected as simultaneously decreased BzCN reactivity and increased DMS reactivity at A47 upon ZMP binding (Extended Data Figure 3f, h). Similarly, in the *crcB* fluoride riboswitch the fluoride-dependent formation of a Hoogsteen base pair between A40 and U48 was detected as simultaneously decreased BzCN reactivity and increased DMS reactivity at A40 (Figure 4f). We therefore suggest that it is useful to perform parallel cotranscriptional RNA structure probing experiments with both BzCN and DMS when possible. DMS probing will likely be advantageous when characterizing long RNA targets due its superior signal-to-noise relative to BzCN probing and when identifying sub-populations in heterogenous mixtures⁴⁹⁻⁵³. In addition, because *in vitro* transcription is typically performed at pH 8.0 (as it is in the current TECprobe procedures), cotranscriptional RNA structure probing experiments are likely to be compatible with four-base DMS probing⁶⁴.”

Minor comments:

The title is very generic – a more specific title making clear that this paper is presenting a new method would be more informative.

We have changed the title to *Observation of coordinated RNA folding events by systematic cotranscriptional RNA structure probing* so that it is clear that the study presents a new high-throughput RNA structure probing method.

TECprobe-ML is somewhat of an awkward name. I recommend not using “ML”, since this will make many readers think of machine learning.

We have changed the name of the method to ‘variable length transcription elongation complex RNA structure probing’ (TECprobe-VL) to avoid confusion.

Page 3: The authors state “This ligation is prone to sequence bias that distorts the reactivity measurement.” Can the authors provide a reference here, as the relevance of bias is controversial.

We now reference Kwok et al., 2013 (A hybridization-based approach for quantitative and low-bias single-stranded DNA ligation) and Poulsen et al., 2015 (SHAPE Selection (SHAPES) enrich for RNA structure signal in SHAPE sequencing-based probing data) following this statement. Both studies found that CircLigase I has an acceptor 3' end bias of dT>dA>>dG>>dC, which agreed with findings made by the manufacturer.

Can the authors provide more detail on the rationale behind the formamide and SSB additions to RT? For example, relevant citations? Are these additions likely to more broadly improve MaP? Also, looking at figure S3b, it is not clear that formamide has an impact relative to SSB.

We have added the text below describing the rationale behind using formamide and SSB, which includes citations that describe the use of these reagents to improve priming specificity and resolve structure in PCR and RT reactions. This text also describes the results of S3b in more detail by emphasizing that, in this experiment, SSB was responsible for resolving primer dimer but that we included 2% formamide in the final procedure because it did not negatively impact the RT reaction and is potentially useful for other RNA targets.

“Formamide and SSB are commonly used to improve primer specificity and relieve secondary structure during PCR and have been used for the same purposes during reverse transcription²⁹⁻³². ET SSB eliminated RT primer dimer, but neither ET SSB nor formamide affected the efficiency at which a ZTP riboswitch RNA was reverse transcribed (Supplementary Figure 3b). Because the inclusion of up to 5% formamide did not inhibit reverse transcription, we chose to include 2% formamide in the reaction, which may relieve secondary structure in some RNA targets.”

Reviewer #2 (Remarks to the Author):

Szyjka and Strobel present an adapted RNA structure probing method for examining the folding landscape of RNAs during transcription called TECprobe-ML. This approach has several advantages over the original co-transcriptional SHAPE-seq: 1) Users can directly amplify the cDNA by priming on a structured 5' cassette instead of ligating an adapter during library preparation. 2) The method now uses mutational profiling which has already been established in other sequencing-based RNA structure probing methods. 3) The authors develop a method to smooth the probing data. The authors show that this approach resolves structures close to the 5' and 3' end of the transcript, which were previously difficult or impossible to detect. The authors use TECprobe-ML to examine the folding pathway of two riboswitches that have been previously characterized by similar methods, the ZTP and fluoride riboswitches, and investigate the folding pathway of the ppGpp riboswitch, which has not been previously characterized. I think the work is thorough and highlights that the method has been optimized from the previous version. I have a few comments:

Thank you for appreciating the thoroughness of the work and how the protocol optimizations have improved cotranscriptional RNA structure probing methodology.

Major comments:

1. It is difficult to evaluate the comparison to the previous iteration of co-transcriptional SHAPE-seq. The authors make a series of claims in the validation sections to show that TECprobe-ML performs better than the original co-transcriptional SHAPE-seq, but it is difficult to determine whether these are supported by their data because there are no direct figure comparisons with the previous work. My concern stems from the fact that the data is different between the two studies and this makes it very difficult for a reader to compare between the two papers. For example, the reactivity (measured by RT stops) values range from 0 – 2 in the Strobel et al. 2019 paper, whereas the reactivities range from 0 – 0.01 in this manuscript (measured by MaP).

Could the authors include a side-by-side comparison of particular nucleotides or transitions with relative reactivities to highlight the improvement over the first method? Are there quantitative metrics that the authors can use to describe how much better TECprobe-ML performs?

We agree that the absence of direct comparisons to previous data was a critical omission in the original manuscript. We considered whether there are quantitative metrics that could be used to compare the data from each approach (e.g. ROC curves as have been used by Weeks and Mustoe), but concluded that given the need to compare data from folding intermediates for which orthogonally determined reference structures do not exist, the clearest comparison would be side-by-side visualization of critical data points. As described above in response to Reviewer 1's first comment, we have addressed this by including seven new supplementary figures in which we directly compare new BzCN probing data with data collected by cotranscriptional SHAPE-Seq. In Supplementary Figures 6 and 10, TECprobe and cotranscriptional-SHAPE-seq reactivity matrices are juxtaposed to show that the overall reactivity trends are the same for both methods. In Supplementary Figures 7, 8, 9, 11, and 12, TECprobe and cotranscriptional-SHAPE-seq reactivity trajectories for specific nucleotides that are referenced in the text are juxtaposed. These figures show that although overall trends are the same for both methods, critical information is either missing or not interpretable in the cotranscriptional SHAPE-Seq data. For example, Supplementary Figure 8c shows that ZMP-dependent decreased reactivity at U94 and G95 of the *pfl* ZTP aptamer, which directly contact ZMP, are not detectable using cotranscriptional SHAPE-Seq. Similarly, Supplementary Figure 11e shows that fluoride-dependent formation of the linchpin base pair, which renders the fluoride aptamer resistant to terminator base pair propagation, was not detectable using cotranscriptional SHAPE-Seq. In these figures, we chose to use actual rather than relative reactivities because this facilitated important comparisons between the methods in some cases. For example, in Supplementary Figure 8b we found that, upon reexamination, the original cotranscriptional SHAPE-Seq data does contain evidence of P3 folding in which the reactivity of G84, C85, and C86 all decrease as P3 folds. However, this was not noticed in the original analysis of these data because the reactivity of C85 and C86 is ~10-fold lower than that of G84. In contrast, the reactivity of these same nucleotides is approximately equivalent when measured by TECprobe-VL. We believe that these new figures provide a thorough comparison of the methods that will enable readers to assess the performance improvements made by TECprobe.

2. In the validation sections, it is sometimes not clear in the text what is new for the TECprobeML and what is already known from previous results. For example, in the ZTP riboswitch section, are the changes seen in the linker between P1/P2 and P3 new results for this method? Similarly, are the coordinated changes in the PK region completely new? How was this detected previously? Please clarify in the text.

We have revised the text to clearly indicate what results are new and what was already known. In cases where a result was known from previous data, we now refer to the relevant supplementary figure in which a direct comparison between TECprobe-VL and cotranscriptional SHAPE-Seq data is made. For example, the paragraph describing changes in the linker that occur upon pseudoknot folding now reads (new text underlined):

“The *pfl* riboswitch pseudoknot comprises base pairs between J1/2 (nts 23-27) and L3 (nts 89-93)²¹ (Figure 3a). ZMP-independent pseudoknot folding was detected as decreased reactivity in J1/2 (BzCN, nts 24-26; DMS, A24, C25) from transcript ~104 to ~110 when nucleotides 89-93 emerge from RNAP (Figure 3f and Extended Data Figure 2e). Pseudoknot folding is coordinated with decreased BzCN reactivity in P2 (nts U31, G32), in the terminal base pair of P1 (A56), and in P1-proximal unpaired nucleotides within the inter-subdomain linker (A77, C78) (Figure 3e).

This indicates that, as expected, pseudoknot folding drives global changes in aptamer structure. While pseudoknot folding was detected in previous data, the coordinated reactivity changes in P1 and the inter-subdomain linker were not³³ (Supplementary Figures 7d and 8a).”

We have made similar additions in all relevant sections of the text. In addition to clarifying what results are new, this also facilitates a systematic comparison of TECprobe-VL and cotranscriptional SHAPE-Seq reactivity trajectories at critical nucleotides.

3. The authors performed full-length structure analysis (TECprobeSL) for the ZTP riboswitch in Supplemental Figure S8. Have the authors done this analysis with different 5' cassette sequences to confirm that it does not interfere with overall folding? I think the functional assays are useful and convincing in this case, but I think it would be useful to show that the full-length structure is also not perturbed by the incorporation of SC1 or other hairpins. This would be important for application to studying the folding of other RNAs which do not have a simple functional assay.

We did test two additional structure cassettes that were longer (to facilitate priming) and which omitted U nucleotides between +3 and +17 so that transcription could be synchronized by initiating transcription without UTP. However, both structure cassettes increased ZMP-independent terminator readthrough and decreased ZMP-dependent antitermination. We now show this result in Supplementary Figure 14. In an unrelated procedure where we also used the SC1 structure cassette (Kelly et al., 2022), we found that transcription could be synchronized using the SC1 leader by positioning an AA dinucleotide downstream of SC1 and initiating transcription without ATP, so this functionality is present in the final leader sequence although we did not use it in the current work.

Additionally, we now include full-length structures that are colored by reactivity for the ZTP and fluoride riboswitches, in which the full-length antiterminated structure is distinct from the terminated structure. We did not include a similar figure for the ppGpp riboswitch because its full-length structure is indistinguishable from the terminated structure. We recognize that this is not a direct comparison to data that were obtained without using a structure cassette and that some RNAs will not be compatible with the inclusion of additional hairpin structures. In the discussion, we now suggest that in these cases it is likely possible to perform a variation of the TECprobe-VL procedure without a 5' structure cassette by using the hybridization-based approach for ssDNA adapter ligation that was developed by Bevilacqua and colleagues (Kwok et al., 2013), which they have used for transcriptome-wide RNA structure probing (Ritchey et al., 2017):

“In cases where a 5' structure cassette cannot be appended to the target RNA, it may be possible to use a hybridization-based strategy to append an adapter to the cDNA 3' end^{26,28}. The main disadvantage of this approach in cotranscriptional RNA structure probing experiments is that it is currently difficult to remove adapter dimer without also depleting short cDNAs.”

We are currently developing a procedure for depleting adapter dimer without any loss of short cDNAs to facilitate TECprobe experiments with RNA targets that are incompatible with a 5' structure cassette. However, the use of a 5' structure cassette will still be highly advantageous when possible because it guarantees that all cDNA is full length so that sequencing coverage is even across the length of the transcript.

4. In the discussion, would the authors expand on why improving resolution near the 3' end is important for studying folding of other RNAs in order to highlight the significance and applications of the improved method.

We have revised the discussion to include the following text, which describes how resolving 3'-proximal structures is crucial for monitoring coordinated structural changes that span the length of a transcript and how this improvement may aid the interpretation of cotranscriptional RNA structure probing data by both manual and automated approaches:

“These limitations are offset by the advantages of using a 5' structure cassette to circumvent the need to ligate an adapter to the cDNA 3' end. Most importantly, this eliminates the need to deplete adapter dimer, which caused poor resolution of 3'-proximal structures in the original cotranscriptional SHAPE-Seq procedure due to the loss of library fragments with short inserts during size selection. In the reanalysis of the ZTP and fluoride riboswitches by TECprobe-VL, resolving 3'-proximal RNA structures enabled the detection of previously unobservable folding events and uncovered new lines of evidence that support known folding events. In the *pfl* ZTP riboswitch system, it became possible to detect the direct interaction of ZMP with U94 and G95 in L3 and the ZMP-dependent stabilization of the G84-C98 base pair, which must be broken for the terminator hairpin to nucleate (Supplementary Figure 8c, d). In addition, the IH1-to-P1 folding transition was detected as both decreased IH1 loop reactivity (previously observed) and decreased downstream P1 stem reactivity (not previously observed) (Supplementary Figure 7a, b). Similarly, in the fluoride riboswitch system, pseudoknot folding was detected as decreased reactivity in both the upstream (previously observed) and downstream (not previously observed) segments of the pseudoknot (Supplementary Figure 11a, b). The ability to detect cotranscriptional folding events as coordinated reactivity changes that span the length of a transcript is likely to aid both manual and automated⁶³ interpretation of cotranscriptional RNA structure probing data.”

5. Would the authors describe how this method compares to other RNA-seq based structure probing methods such as SPET-seq?

We have revised the discussion to include a paragraph that describes some similarities and differences between TECprobe-VL and SPET-seq (included here below). While the overall goal of TECprobe and SPET-seq is the same, there are some fundamental differences at the level of *in vitro* transcription. As we describe below, like cotranscriptional SHAPE-Seq, TECprobe distributes transcription elongation complexes across DNA using biotin-streptavidin roadblocks which halt RNAP efficiently enough to permit transcription to be performed as a single-round reaction. SPET-seq uses an orthogonal approach in which multi-round transcription is halted by adding the intercalating agent Actinomycin D and DNaseI to the reaction immediately before structure probing and subsequently isolating intermediate transcripts by gel extraction. Importantly, the optimizations we have made at the levels of library prep and data analysis are generalizable to all cotranscriptional RNA structure probing methods, including SPET-seq.

“Our validation and benchmarking of TECprobe-VL focused on comparisons with Cotranscriptional SHAPE-Seq because the two methods use closely related *in vitro* transcription procedures in which RNAP is halted at biotin-streptavidin roadblocks under single-round conditions. *In vitro* cotranscriptional RNA structure probing has also been performed using an orthogonal procedure, called Structural Probing of Elongating Transcripts (SPET-seq)¹⁶. In an *in vitro* SPET-seq experiment, a multi-round *in vitro* transcription reaction is halted by the simultaneous addition of actinomycin D and DNaseI before nascent RNA is chemically probed.

The resulting intermediate transcripts are then further enriched by gel extraction before conversion into a sequencing library. We prefer to use template DNA strand biotin-streptavidin complexes to distribute RNA polymerase across the DNA template because they halt transcription with >80% efficiency and the resulting complexes are >99% stable¹⁵. This enables the use of single-round *in vitro* transcription, which simplifies the transcription reaction. However, in cases where multi-round transcription conditions are advantageous or preferred, all of the sequencing library construction optimizations that we have described are generalizable to SPET-seq. Furthermore, the use of neighboring transcript smoothing to address uneven intermediate transcript representation is generalizable to all cotranscriptional RNA structure probing methods.”

6. What is the outlook for expanding upon this method? Can this approach be combined with other MaP-based structure probing approaches that examine RNA structural changes upon protein binding or examine the structural heterogeneity?

Thank you for this suggestion. One of our goals for developing TECprobe-VL was to establish a procedure that would be a foundation for more demanding applications of cotranscriptional RNA structure probing. We have revised the discussion at three points to emphasize this:

In the first paragraph we state that the use of MaP “enables the possibility of integrating TECprobe-VL with MaP-based methods for resolving structural heterogeneity⁴⁹⁻⁵³ and detecting RNA-protein interactions⁵⁴”

Following a discussion of the utility of parallel BzCN and DMS probing we state “In addition, because *in vitro* transcription is typically performed at pH 8.0 (as it is in the current TECprobe procedures), cotranscriptional RNA structure probing experiments are likely to be compatible with four-base DMS probing⁶⁴.”

And in the final sentence we state “Furthermore, the high efficiency of TECprobe-VL will likely enable its use as the basis for developing new cotranscriptional RNA structure probing procedures that overcome the limitations of current method.”

We have begun building upon the TECprobe platform to enable more demanding applications of cotranscriptional RNA structure probing, however this work is too preliminary to discuss in the current manuscript.

Minor comments:

1. With regards to performance, in Figure 2a – what are the alignment rates for the original cotranscriptional SHAPE-seq?

The alignment rate for the original cotranscriptional SHAPE-Seq procedure ranged from 20% to 60% depending on how much adapter dimer was present in the library. We now state this in the sentence :

“Because this size selection procedure attempted to retain short cDNA products, the removal of adapter dimer was often incomplete, which caused alignment rates of 20% to 60%, depending on the abundance of adapter dimer in the sequencing library.”

2. Figure 2b-d – Why are there large regions in all three cases that are not enriched with the biotin-streptavidin roadblocks? How is this determined?

Thank you for pointing out that we had not clearly explained this. We have added the following text to the results, which explains that the regions that are not enriched for intermediate transcripts are caused by the lack of template DNA strand biotin-streptavidin roadblocks in the reverse primer used for PCR. Although there are non-template DNA strand biotin-streptavidin complexes at these positions, they do not efficiently halt RNAP.

“Like Cotranscriptional SHAPE-Seq, TECprobe-VL experiments begin by distributing TECs across template DNA in a single-round *in vitro* transcription reaction using biotin-streptavidin roadblocks, which halt RNAP ~10 nts upstream of the biotin site, so that cotranscriptionally displayed intermediate transcripts can be chemically probed¹⁵. Internal biotin modifications are incorporated into template DNA by including biotin-11-dNTPs during PCR and a terminal biotin modification that prevents any run-off transcription with >99% efficiency²⁴ is included in the reverse primer used for PCR. Internal biotin-streptavidin roadblocks halt transcription more efficiently when they are positioned in the template DNA strand (>80%) than in the non-template DNA strand (~30%)¹⁵. Consequently, DNA templates that are structured as described above enrich for intermediate transcripts from the transcription start site until ~10 nucleotides before the segment of template DNA that is composed of the reverse primer and at a cluster of positions ~10 nucleotides upstream of the terminal biotin-streptavidin complex (Figure 2b-d, Extended Data Figure 1a-c, top plots).”

3. Can the authors comment on whether the SC1 cassette is needed if the addition of ET SSB solves the primer dimer issue as in Figure S3b? Can you add SSB to the cDNA amplification step?

In this case, the SC1 structure cassette is still needed because it is used as a constant priming site during PCR, which circumvents the need to ligate an adapter to the cDNA 3' end and enriches for full-length cDNA. We have not tried adding SSB during cDNA amplification since we have not observed issues with primer dimer, however it is certainly possible to do and SSB is known to improve the primer specificity and DNA polymerase processivity. As described above in response to Reviewer 1's last minor comment, we have added text to the results that references prior work in which the utility of SSB (and formamide) for PCR and reverse transcription was described.

4. The authors should add the descriptions of the color coding (i.e. in the ZTP riboswitch section, green/purple for 0 and 1 mM ZMP) to all the figure legends for clarity.

We have revised all relevant figure legends to include this information.

5. For the ppGpp riboswitch, the authors should include a secondary structure diagram of the complete ppGpp riboswitch they tested (transcript 134 I believe). They discuss late folding events that occur with transcripts at 108 and 115 nts but there is no complete structure shown in Figure 7 or Figure S13.

We have added two figures showing the complete ppGpp riboswitch. In Extended Data Figure 9c, we show the termination product that occurs at transcript 134 colored by the DMS reactivity of the -ppGpp condition. In Extended Data Figure 9g we show the folded ppGpp aptamer in the presence of a hairpin that is predicted to fold within the downstream terminator stem. The

existence of this structure in the +ppGpp folding pathway cannot be assessed by cotranscriptional RNA structure probing because the full-length riboswitch adopts a terminated structure by the time chemical probing is performed.

6. In Figure 6k and l, I think the labels for P2a and P2b are shifted. P2a should be near the base of the coaxial stack.

Thank you for catching this error. We have fixed this and check all other relevant figures.

Reviewer #3 (Remarks to the Author):

RNAs fold during transcription and this is important for their regulation because the timing of hairpin formation can regulate the function of genes. In this manuscript, Szyjka and Strobel develop and optimize a method for probing co-transcriptional folding that they call “TECprobe-ML”. This study represents both method development and key biological insights into principles of RNA folding and riboswitch gene regulation. TECprobe-ML is an improvement over their previous method of “Co-transcriptional SHAPE-seq” and they go on to clearly list all the improvements. The authors probe the folding of a set of riboswitches using this method, first validating it on the ZTP and fluoride riboswitches and then extending it to the ppGpp riboswitch. Validation sections, which are critical parts of any methods development section of a manuscript, read very well and provide new insights, including information on 3'-proximal residues and read-through of very stable RNA structures such as terminators. There are nice controls that addition of a 5'-element does not affect response of antitermination to ligand. Results on the three riboswitches support formation of key helices in antitermination. This is an important paper because it can aid modeling and prediction of co-transcriptional folding of RNA, which is understudied and not well-understood but is prevalent and dominates the folding of many RNAs. I ask the authors to address the following points.

Thank you for appreciating the potential utility of this work for aiding the prediction of cotranscriptional RNA folding and the importance of the validation sections of the manuscript.

1. Having “ML” in a method title immediately implies machine learning. The authors intend for it to stand for “multilength”. I strongly encourage them to choose a different abbreviation to avoid confusion.

We have changed the name of the method to ‘variable length transcription elongation complex RNA structure probing’ (TECprobe-VL) to avoid confusion.

2. The first section of the Results, which explains the TECprobe-ML, is missing any description of the “biotin-streptavidin” roadblock depicted in Figure 1. Presumably this roadblock got introduced (randomly?) in solid-phase synthesis of the BS DNA or during a PCR of the dsDNA template. Ref 15 suggests that the PI developed this in the Lucks lab but at least some description is needed here.

Thank you for pointing out that we had not clearly explained the transcription roadblocking strategy. We have added the following text to the beginning of the results to explain how the biotin modifications are incorporated into the DNA template and how the roadblocks enrich for intermediate transcripts:

“Like Cotranscriptional SHAPE-Seq, TECprobe-VL experiments begin by distributing TECs across template DNA in a single-round *in vitro* transcription reaction using biotin-streptavidin roadblocks, which halt RNAP ~10 nts upstream of the biotin site, so that cotranscriptionally displayed intermediate transcripts can be chemically probed¹⁵. Internal biotin modifications are incorporated into template DNA by including biotin-11-dNTPs during PCR and a terminal biotin modification that prevents any run-off transcription with >99% efficiency²⁴ is included in the reverse primer used for PCR. Internal biotin-streptavidin roadblocks halt transcription more efficiently when they are positioned in the template DNA strand (>80%) than in the non-template DNA strand (~30%)¹⁵. Consequently, DNA templates that are structured as described above enrich for intermediate transcripts from the transcription start site until ~10 nucleotides before the segment of template DNA that is composed of the reverse primer and at a cluster of positions ~10 nucleotides upstream of the terminal biotin-streptavidin complex (Figure 2b-d, Extended Data Figure 1a-c, top plots).”

3. Data smoothing across three lengths plays an important part in enhancing replication of the data. This is a purely computational change, however, and so one wonders if smoothing might have helped the original non-MaP approach.

We agree that data smoothing would likely have improved the original Cotranscriptional SHAPE-Seq approach. However, one of the main limitations of the original method was the loss of information at 3'-proximal nucleotides, which smoothing would not have been able to recover. This can be seen clearly in several new supplementary figures in which Cotranscriptional SHAPE-Seq and TECprobe-VL data are compared. For example, in Supplementary Figure 8c, a ZMP-dependent decrease in the reactivity of U94 and G95, which directly contact ZMP, is clearly observed by TECprobe-VL whereas nothing can be concluded about these nucleotides from the original cotranscriptional SHAPE-Seq data.

Within the discussion, we now state that “Furthermore, the use of neighboring transcript smoothing to address uneven intermediate transcript representation is generalizable to all cotranscriptional RNA structure probing methods.”

4. Weeks and Mustoe have turned to DMS mapping over SHAPE, at least for PAIR-MaP that uses single-molecule correlated chemical probing. This is because DMS is much more sensitive than SHAPE for mutational profiling, as established by Rouskin. This requires pH 8 to hit all four bases, but transcription is often fine at this pH. Have the authors considered switching to DMS-based method over SHAPE? Clearly, they can use DMS, as evidenced by SI figures with DMS probing for A/C on the ZTP and fluoride riboswitches.

As described above in response to Reviewer 1, we revised the discussion to describe the utility and performance of BzCN and DMS probing in cotranscriptional RNA structure probing experiments. Our current recommendation is to use both whenever possible, however we emphasize the advantages of DMS probing in the following sentences:

“DMS probing will likely be advantageous when characterizing long RNA targets due its superior signal-to-noise relative to BzCN probing and when identifying sub-populations in heterogenous mixtures⁴⁹⁻⁵³. In addition, because *in vitro* transcription is typically performed at pH 8.0 (as it is in the current TECprobe procedures), cotranscriptional RNA structure probing experiments are likely to be compatible with four-base DMS probing⁶⁴.”

5. p6. J1/2. It would be helpful to annotate Fig 3e with J1/2 on the appropriate nucleotides on the x-axis as was done for other elements. The authors should check all their writing and figures for such omissions and put in annotations to help the reader.

Thank you for pointing out this omission. We have added this annotation, have checked for other omissions, and have added annotations to plots throughout the manuscript to improve their readability.

6. p9. It is interesting and at times convincing to read about the intermediate hairpins (IH) that precede formation of P2 in the ppGpp riboswitch. Given that P2 is a long element, this is not surprising as it is hard to imagine it otherwise given RNA's proclivity to fold; nonetheless, it is impressive to see. This raises a deeper question: Do IH hairpins serve any purpose in guiding folding? For example, is there evidence for their conservation when doing sequence comparison across ppGpp riboswitches? See papers from Isambert and Siggia where such "folding guides" have been proposed for such His: Isambert, H. & Siggia, E. D. (2000). Modeling RNA folding paths with pseudoknots: Application to hepatitis delta virus ribozyme. Proc. Natl. Acad. Sci. U S A 97, 6515-6520. 18642

Thank you for suggesting that we investigate the potential purpose of intermediate hairpins further. To assess this, we identified 30 putative ppGpp riboswitches within the initial analysis of ykkC-variant riboswitches by Nelson et al., (2017) and used minimum free energy structure prediction to assess whether these sequences have the capacity to form intermediate hairpins prior to P2 folding. The results of this analysis are presented in Supplementary Figure 13 and described in the following text:

"We assessed whether intermediate structures like IH1, IH2, and IH3 are present in other ppGpp riboswitches by predicting minimum free energy structures using the RNAstructure⁴⁸ Fold command. 18 of the 30 putative ppGpp aptamers that were assessed were predicted to form an IH1-like structure in which a conserved YCU trinucleotide base pairs with conserved purines at the 5' end of the aptamer (Supplementary Figure 13a-c, f). Of these structures, 80% have a minimum free energy that is less than or equal to that of the experimentally verified *Cba* IH1 structure (Supplementary Figure 13e). IH1 is therefore a consequence of two closely positioned ppGpp aptamer sequence elements having a high likelihood of forming contiguous base pairs. All ppGpp aptamers are predicted to form some non-native structure when the transcript is sufficiently long to form the IH2 and IH3 hairpins observed in the *Cba* ppGpp aptamer (Supplementary Figure 13g). However, these predicted structures are heterogenous, which is expected because they are composed of base pairs between conserved and variable regions of the aptamer (Supplementary Figure 13 c, d, h)."

We have also added the paragraph below to the discussion, which discusses the potential implications of the observation that non-native structures precede folding of native ZTP and ppGpp aptamer structures. We agree that it is not surprising that transient non-native structures tend to fold when there is sufficient distance between nucleotides that form native base pairs. In the discussion paragraph noted above, we suggest that efficient resolution of non-native structures is likely to be particularly important for riboswitches that regulate transcription because the window of time for aptamer folding is limited.

"Analysis of ppGpp riboswitch folding using TECprobe-VL established a positional map of folding intermediates that can occur as the *Cba* ppGpp aptamer folds (Figure 8). Like the *pfl* ZTP aptamer, transient intermediate hairpins precede *Cba* ppGpp aptamer folding. The first

transient structure, IH1, is potentiated by the high likelihood of forming contiguous base pairs between conserved poly-purine and poly-pyrimidine stretches within the ppGpp aptamer and therefore may be present in other ppGpp riboswitch folding pathways. Minimum free energy structure predictions suggest that the subsequent transient structures that were observed for the *Cba* ppGpp riboswitch, IH2 and IH3, are not present in other ppGpp riboswitch folding pathways (Supplementary Figure 13). However, all putative ppGpp riboswitches that were assessed are predicted to form non-native structures that precede P2 hairpin folding. Determining whether the observed transient structures function as ‘folding guides’⁶⁶ that coordinate *Cba* ppGpp aptamer folding is not possible using the current data and will likely require approaches for evaluating how perturbing these structures affects riboswitch function. In addition to the *pfl* ZTP and *Cba* ppGpp riboswitches, transient non-native structures have also been detected within the folding pathways of the *E. coli* signal recognition particle, RNase P, tmRNA, and *thiB* riboswitch RNAs^{14,16,67,68}. In most of these cases, the non-native structure precedes folding of a native structure that requires base pairs between distal nucleotides. While some transient structures are conserved^{56,61}, others, such as the *pfl* ZTP riboswitch IH1 hairpin and *Cba* ppGpp riboswitch IH2 and IH3 hairpins, appear to be a consequence of the nascent transcript sampling energetically favorable structures until it is long enough that the native structure can fold. The occurrence of non-native structures in such contexts is not surprising given the capacity of random RNA sequences for the formation of secondary structures^{69,70}. In such cases, intermediate structures that can be easily resolved to permit the native structure to fold are likely advantageous. This is likely to be particularly important for riboswitches that regulate transcription because the resolution of non-native structures must occur cotranscriptionally.”

7. p9, para 2. There is a difference between equilibrated and co-transcriptional IH data with BzCN. How much would this change with pausing and native polymerases. (The authors use *E. coli* polymerase for a Clostridiales bacterium riboswitch.)

This is an excellent question. Although we cannot be certain with the current data, we suspect that these factors would not change this outcome for the following reasons: The outcome of cotranscriptional RNA structure probing experiments is expected to be sensitive to transcription pauses because pausing can still influence an RNA folding trajectory before RNAP stalls at a roadblock. It is possible for nascent RNA structure to equilibrate to some degree while RNAP is stalled, however this cannot be circumvented because it is necessary to stop transcription before chemical probing is performed. We do not know how the transcription pausing properties or RNAP-nascent RNA interactions of *E. coli* RNAP differ from those of the cognate polymerase. However, we anticipate that the high efficiency of TECprobe-VL will begin to enable experiments using other RNA polymerases which may not be as efficient as *E. coli* RNAP *in vitro*. We have revised the final paragraph of the discussion to include a description of these limitations in the following sentences:

“Third, the current method uses *E. coli* RNAP to transcribe RNA regardless of the origin of the target RNA. This could potentially influence RNA folding outcomes if the transcription pausing properties or RNAP-nascent RNA interactions of *E. coli* RNAP are different than those of the cognate RNAP³⁻⁵.”

8. Figure 7a could be clearer. What are the dashed boxes for in 7a? Is this the “long-range interactions with the minor group of P2b”? Annotate in both 7a and 7b. Specifically, in 7b, annotate the various pairings. Other figures should be cross-checked for similar changes.

We have revised Figure 7 (and all related figures) to indicate that the connected dashed boxes refer to the long-range contacts between P2b and P4. We have also annotated these interactions and the location of P1 in 7b.

9. p11. The authors write, “In contrast, ppGpp-dependent reactivity signatures were not detectable in transcripts that accumulated at the terminal transcription roadblock downstream of the termination site (Figure 7 and Supplementary Figure 13h-l).” In Supp Fig 13h-l, do they want us to look at the little strip at 170? If so, it should be stated. I can see ppGpp-dependent reactivity signatures in transcripts downstream of termination for lengths 126-135 in this figure.

Thank you for pointing out that this was not clear. We did intend to refer to the transcript lengths from 169 to 172 and have now made this clear in the text by changing the figure call out to “Figure 7 and Extended Data Figure 9f, nucleotides 169 to 172”. We have also revised other relevant figure call outs in the same way.

Additional changes

In addition to the revisions described in the response above, we have made several other changes to the manuscript which we describe below.

- The annotation for the ppGpp binding window on the reactivity matrices has been corrected so that it matches the annotations in the reactivity trajectory plots.
- The ppGpp riboswitch terminator hairpin is now drawn with an extended helix in which the poly-U tract base pairs with the poly-purine tract that spans nucleotides 76 to 84 of the ppGpp aptamer, which is consistent with BzCN and DMS reactivity data.
- We identified a third possible intermediate hairpin (IH3) that is consistent with cotranscriptional BzCN and DMS probing data and have revised Figure 6 and the relevant section of the text accordingly.

REVIEWERS' COMMENTS

Reviewer #1 (Remarks to the Author):

Strobel and colleagues have done a good job addressing reviewer critiques. Important additions include comparisons with data from prior methods, which clearly establish the improved quality of their method. Additional analyses of the ppGpp riboswitch system, figure reorganization, and expanded discussion also help clarify the novel findings of their experiments.

Despite some streamlining, the manuscript is still quite dense, and the accumulation of separate "extended" and "supporting" figures makes it challenging to follow in places. At least to this reader, it would be easier to closely inspect several well-annotated matrices rather than having to bounce between 100s of individual nt traces across 3 locations. Further streamlining the text and figures would make the manuscript more accessible to the average reader.

Several other specific critiques:

-I continue to find the layout of Figure 7 awkward. The mix of nts in panels f/g is confusing. It would also facilitate comparisons to show the same intermediates (e.g. a/h) adjacent to each other rather than at the top and bottom of the figure.

-Line 475: authors should remove the comment about "a fourth riboswitch that we are currently characterizing".

-A description of how the co-existing IH3 state was identified/modeled should be provided.

-A greater description of how reactivity data were used to guide structure modeling of ppGpp riboswitch in methods is needed.

Reviewer #2 (Remarks to the Author):

The authors have added a number of supplemental figures to facilitate direct comparisons of their new method with the previous method which adds to the impact of the work. All of my comments have been sufficiently addressed in the revised manuscript.

Reviewer #3 (Remarks to the Author):

I am satisfied with the authors' responses to my concerns. I also looked over the main concerns of Reviewers 2 and 3 and feel that they did a good job here as well. I think this is an impactful paper that is appropriate for Nature Comm.

REVIEWERS' COMMENTS

Reviewer #1 (Remarks to the Author):

Strobel and colleagues have done a good job addressing reviewer critiques. Important additions include comparisons with data from prior methods, which clearly establish the improved quality of their method. Additional analyses of the ppGpp riboswitch system, figure reorganization, and expanded discussion also help clarify the novel findings of their experiments.

Despite some streamlining, the manuscript is still quite dense, and the accumulation of separate "extended" and "supporting" figures makes it challenging to follow in places. At least to this reader, it would be easier to closely inspect several well-annotated matrices rather than having to bounce between 100s of individual nt traces across 3 locations. Further streamlining the text and figures would make the manuscript more accessible to the average reader.

Thank you for suggesting that we consider ways in which we could further streamline the manuscript.

While we recognize that the number of nucleotide traces remains high, we believe that it is important to keep these plots in the manuscript to ensure that the data are presented as rigorously as possible. While the reactivity matrices are useful for visualizing whole data sets, they have several limitations: First, the color scale used for the heatmaps is capped for presentation purposes, so the entire range of reactivity values is not shown in the heatmap. Second, it can be difficult to perceive reactivity values quantitatively when they are presented as a heatmap. Third, in some cases biologically meaningful changes in the reactivity of different nucleotides occur at different scales and therefore cannot be visualized clearly in a single heatmap. The nucleotide trace plots overcome these limitations and are, in our opinion, important for the presentation of these data.

To further streamline the manuscript, we have removed text in which reactivity changes in specific nucleotides were called out parenthetically, which interrupted the flow of sentences and made some sections of the text difficult to read. In the revised text, the description of folding events focuses on specific structures (e.g. P1, L1, etc.) rather than individual nucleotides whenever possible. In the figures, plots are labeled both by the relevant structure and the nucleotides that are plotted, so this information remains available in a much less obtrusive way. Although this is a small change, we believe that it substantially improves the readability and accessibility of the text, and we have provided an example of one section below:

Previous text:

P1 folding was detected as decreased reactivity in the IH1 loop (BzCN, U16, A17; DMS, nts 15-17) and the downstream P1 stem (BzCN, nts 44-47, U49, A50, A56; DMS, A50, A53, C54) from transcript ~67 to ~80, as the downstream P1 stem (nts 44-56) emerges from RNAP (Figure 3a, e and Supplementary Figure 8c). P2 folding was detected at transcript ~60 as decreased P2 reactivity (BzCN, U31, G32; DMS, nts 39-41) when the entire P2 hairpin has emerged from RNAP (Figure 3e and Supplementary Figure 8d).

Current text:

P1 folding was detected as decreased reactivity in the IH1 loop and the downstream P1 stem from transcript ~67 to ~80, as the downstream P1 stem emerges from RNAP (Figure 3a, e and Supplementary Figure 8c). P2 folding was detected at transcript ~60 as decreased P2 reactivity when the entire P2 hairpin has emerged from RNAP (Figure 3e and Supplementary Figure 8d).

Several other specific critiques:

-I continue to find the layout of Figure 7 awkward. The mix of nts in panels f/g is confusing. It would also facilitate comparisons to show the same intermediates (e.g. a/h) adjacent to each other rather than at the top and bottom of the figure.

Thank you for suggesting that we consider how the visualization of ppGpp riboswitch folding intermediates could be improved. We have made the following changes to improve the presentation of these findings: First, as suggested, we have placed the secondary structures adjacent to each other to facilitate comparisons between the BzCN and DMS reactivity profiles that support these structures. Second, we have reorganized the nucleotide traces so that they are grouped by the folding event that they support (i.e. P2c, P2b, or P2a folding). This arrangement of nucleotide traces mirrors how these data are discussed and therefore simplifies this section of the text. These changes have substantially improved the clarity of these data and we appreciate the reviewer's suggestion that led to these revisions.

-Line 475: authors should remove the comment about "a fourth riboswitch that we are currently characterizing".

We have removed this section of the sentence.

-A description of how the co-existing IH3 state was identified/modeled should be provided.

Prior to the initial submission of the manuscript, we had identified possible folding intermediates by predicting the structure of ppGpp riboswitch substrings using the RNAstructure Fold command. The set of intermediate structures was then manually curated based on the agreement of the predicted structure with reactivity data and by assessing which nucleotides had emerged from RNAP when the intermediate structure was observed and when the intermediate rearranged into a subsequent structure. The IH1 and IH2 hairpins, which are clearly detectable in the equilibrium BzCN probing data, were identified using this approach. However, IH3, which does not appear to be present in the equilibrium BzCN probing data, was overlooked. During revision, we used the RNAstructure stochastic command to assess possible base pairing configurations for entire transcripts (excluding RNA that would be present within the RNAP footprint). This approach revealed the IH3 structure, which is consistent with cotranscriptional BzCN and DMS probing data and may explain why nucleotides 33-35 within IH2b are moderately reactive. We have added a description of how intermediate ppGpp aptamer structures (including IH3) and the apo and holo ppGpp aptamer were modeled and have included this text below in response to the next comment.

-A greater description of how reactivity data were used to guide structure modeling of ppGpp riboswitch in methods is needed.

We have revised the methods to include the following text, which describes how structure modeling was performed:

Modeling and visualization of RNA secondary and tertiary structures

Non-native intermediate ppGpp riboswitch structures were modeled as follows: For each transcript of interest, the 3'-most 12 to 14 nt were forced to be single-stranded to account for the *E. coli* RNAP footprint and possible base pairing configurations for the remaining transcript were generated using the RNAstructure v6.4⁴⁹ stochastic command with default settings. From this set of structures, putative folding intermediates were manually selected based on the agreement

of the predicted structure with reactivity data and by assessing which nucleotides had recently emerged from RNAP at the transcript length when the intermediate is first observed and when the intermediate rearranges into a subsequent structure. The holo ppGpp aptamer structure was modeled by manually assessing the agreement of holo ppGpp aptamer reactivity profiles with conserved secondary and tertiary structure elements that were identified crystal structures^{47,48}. The apo ppGpp aptamer, for which a crystal structure does not exist, was modeled by identifying ppGpp-dependent differences between the apo and holo ppGpp aptamer reactivity profiles and determining the interactions that these nucleotides make within the crystal structures; in all cases, DMS reactivity increased in the absence of ppGpp which indicates exposure of the Watson-Crick face and implies that a pairing interaction observed in the holo aptamer is either transient or nonexistent in the apo aptamer. Crystal structures were visualized using UCSF Chimera⁸⁰.

Reviewer #2 (Remarks to the Author):

The authors have added a number of supplemental figures to facilitate direct comparisons of their new method with the previous method which adds to the impact of the work. All of my comments have been sufficiently addressed in the revised manuscript.

Reviewer #3 (Remarks to the Author):

I am satisfied with the authors' responses to my concerns. I also looked over the main concerns of Reviewers 2 and 3 and feel that they did a good job here as well. I think this is an impactful paper that is appropriate for Nature Comm.